# TacticAI: an AI assistant for football tactics

Zhe Wang ●[1,6] ✉, Petar Veličković ●[1,6] ✉, Daniel Hennes ●[1,6],
Nenad Tomašev ●[1], Laurel Prince[1], Michael Kaisers[1], Yoram Bachrach[1],
Romuald Elie[1], Li Kevin Wenliang[1], Federico Piccinini[1], William Spearman[2],
Ian Graham[3], Jerome Connor[1], Yi Yang[1], Adrià Recasens[1], Mina Khan[1],
Nathalie Beauguerlange[1], Pablo Sprechmann[1], Pol Moreno[1], Nicolas Heess ●[1],
Michael Bowling ●[4], Demis Hassabis[1] & Karl Tuyls ●[5] ✉

Identifying key patterns of tactics implemented by rival teams, and developing effective responses, lies at the heart of modern football. However, doing so algorithmically remains an open research challenge. To address this unmet need, we propose TacticAI, an AI football tactics assistant developed and evaluated in close collaboration with domain experts from Liverpool FC. We focus on analysing corner kicks, as they offer coaches the most direct opportunities for interventions and improvements. TacticAI incorporates both a predictive and a generative component, allowing the coaches to effectively sample and explore alternative player setups for each corner kick routine and to select those with the highest predicted likelihood of success. We validate TacticAI on a number of relevant benchmark tasks: predicting receivers and shot attempts and recommending player position adjustments. The utility of TacticAI is validated by a qualitative study conducted with football domain experts at Liverpool FC. We show that TacticAI's model suggestions are not only indistinguishable from real tactics, but also favoured over existing tactics 90% of the time, and that TacticAI offers an effective corner kick retrieval system. TacticAI achieves these results despite the limited availability of gold-standard data, achieving data efficiency through geometric deep learning.

Association football, or simply football or soccer, is a widely popular and highly professionalised sport, in which two teams compete to score goals against each other. As each football team comprises up to 11 active players at all times and takes place on a very large pitch (also known as a soccer field), scoring goals tends to require a significant degree of strategic team-play. Under the rules codified in the Laws of the Game[1], this competition has nurtured an evolution of nuanced strategies and tactics, culminating in modern professional football leagues. In today's play, data-driven insights are a key driver in determining the optimal player setups for each game and developing counter-tactics to maximise the chances of success[2].

When competing at the highest level the margins are incredibly tight, and it is increasingly important to be able to capitalise on any opportunity for creating an advantage on the pitch. To that end, top-tier clubs employ diverse teams of coaches, analysts and experts, tasked with studying and devising (counter-)tactics before each game. Several recent methods attempt to improve tactical coaching and player decision-making through artificial intelligence (AI) tools, using a wide variety of data types from videos to tracking sensors and applying diverse algorithms ranging from simple logistic regression to elaborate neural network architectures. Such methods have been employed to help predict shot events from videos[3], forecast off-screen movement from spatio-temporal data[4], determine whether a match is in-play or interrupted[5], or identify player actions[6].

The execution of agreed-upon plans by players on the pitch is highly dynamic and imperfect, depending on numerous factors

[1]Google DeepMind, 6-8 Handyside Street, London N1C 4UZ, UK. [2]Liverpool FC, AXA Training Centre, Simonswood Lane, Kirkby, Liverpool L33 5XB, UK.
[3]Liverpool FC, Kirkby, UK. [4]University of Alberta, Amii, Edmonton, AB T6G 2E8, Canada. [5]Google DeepMind, London, UK. [6]These authors contributed equally:
Zhe Wang, Petar Veličković, Daniel Hennes. ✉e-mail: zhewang@google.com; petarv@google.com; ktuyls@gmail.com

including player fitness and fatigue, variations in player movement and positioning, weather, the state of the pitch, and the reaction of the opposing team. In contrast, set pieces provide an opportunity to exert more control on the outcome, as the brief interruption in play allows the players to reposition according to one of the practiced and pre-agreed patterns, and make a deliberate attempt towards the goal. Examples of such set pieces include free kicks, corner kicks, goal kicks, throw-ins, and penalties[2].

Among set pieces, corner kicks are of particular importance, as an improvement in corner kick execution may substantially modify game outcomes, and they lend themselves to principled, tactical and detailed analysis. This is because corner kicks tend to occur frequently in football matches (with ~10 corners on average taking place in each match[7]), they are taken from a fixed, rigid position, and they offer an immediate opportunity for scoring a goal—no other set piece simultaneously satisfies all of the above. In practice, corner kick routines are determined well ahead of each match, taking into account the strengths and weaknesses of the opposing team and their typical tactical deployment. It is for this reason that we focus on corner kick

analysis in particular, and propose TacticAI, an AI football assistant for supporting the human expert with set piece analysis, and the development and improvement of corner kick routines.

TacticAI is rooted in learning efficient representations of corner kick tactics from raw, spatio-temporal player tracking data. It makes efficient use of this data by representing each corner kick situation as a graph—a natural representation for modelling relationships between players (Fig. 1A, Table 2), and these player relationships may be of higher importance than the absolute distances between them on the pitch[8]. Such a graph input is a natural candidate for graph machine learning models[9], which we employ within TacticAI to obtain high-dimensional latent player representations. In the Supplementary Discussion section, we carefully contrast TacticAI against prior art in the area.

Uniquely, TacticAI takes advantage of geometric deep learning[10] to explicitly produce player representations that respect several symmetries of the football pitch (Fig. 1B). As an illustrative example, we can usually safely assume that under a horizontal or vertical reflection of the pitch state, the game situation is equivalent. Geometric deep

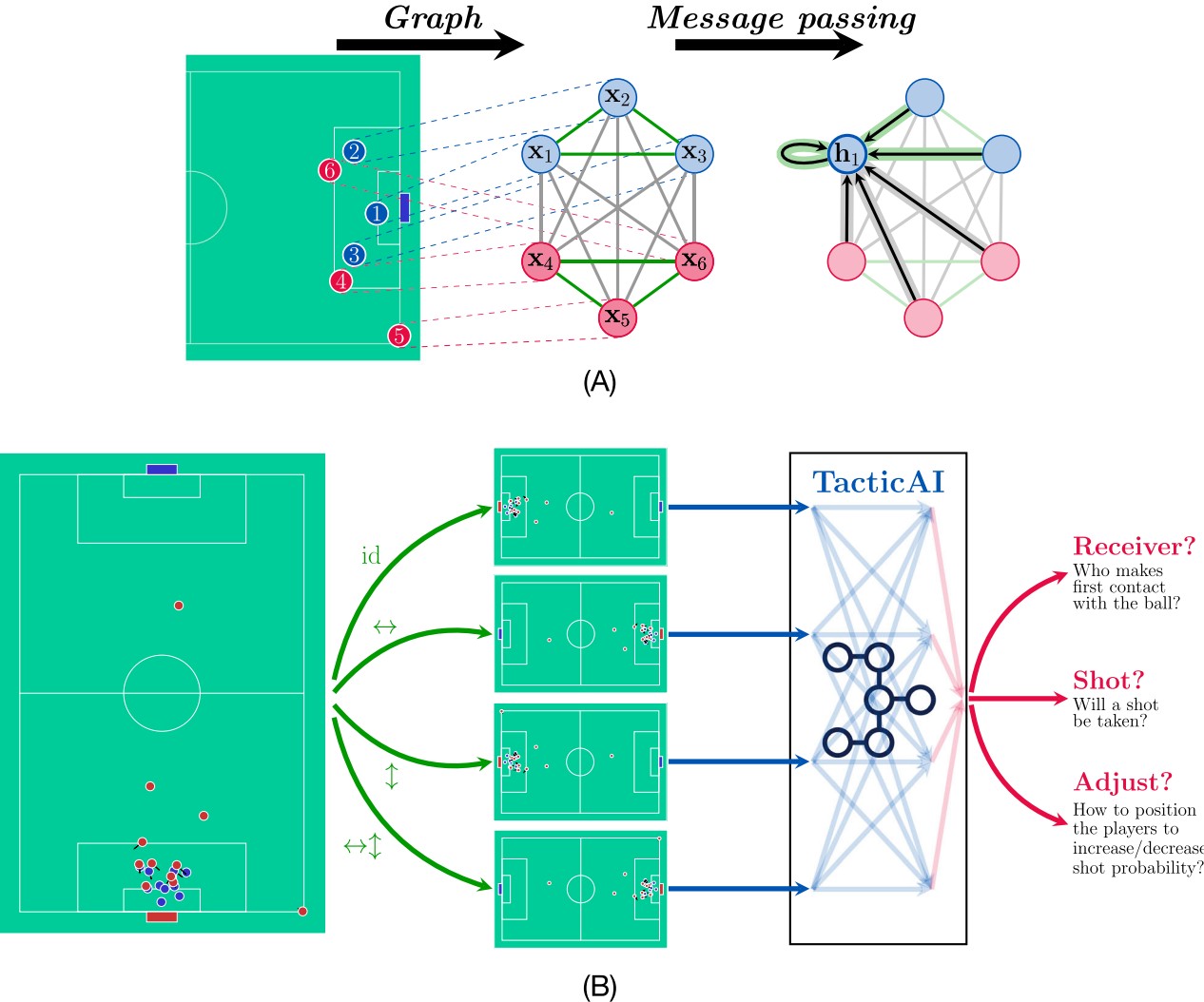

**Fig. 1 | A bird's eye overview of TacticAI. A** How corner kick situations are converted to a graph representation. Each player is treated as a node in a graph, with node, edge and graph features extracted as detailed in the main text. Then, a graph neural network operates over this graph by performing message passing; each node's representation is updated using the messages sent to it from its neighbouring nodes. **B** How TacticAI processes a given corner kick. To ensure that TacticAI's answers are robust in the face of horizontal or vertical reflections, all possible combinations of reflections are applied to the input corner, and these four views are then fed to the core TacticAI model, where they are able to interact with each other to compute the final player representations—each internal blue arrow corresponds to a single message passing layer from (**A**). Once player representations are computed, they can be used to predict the corner's receiver, whether a shot has been taken, as well as assistive adjustments to player positions and velocities, which increase or decrease the probability of a shot being taken.

**Table 1 | Summary of the features used in the corresponding tasks**

| Benchmark task | Node features | Edge features | Global features |
|---|---|---|---|
| Receiver prediction (Node classification) | Player positions | Teammate or opponent | None |
| | Player velocities | | |
| | Player weights | | |
| | Player heights | | |
| | Ball possession | | |
| Shot prediction (Graph classification) | Same as above | Same as above | Receiver ID |
| Guided generation (Node regression) | Same as above | Same as above | Shot indicator |
| | Receiver ID | | |

**Table 2 | Summary of the details of the features used to construct graphs**

| Feature | Feature type | Explanation |
|---|---|---|
| Player positions | Node | *XY*-positions of 22 players on the pitch. |
| Player velocities | Node | *XY*-velocities of 22 players on the pitch. |
| Player weights | Node | Weights of 22 players. |
| Player heights | Node | Heights of 22 players. |
| Ball possession | Node | Binary indicator to indicate whether this player is possessing the ball. |
| Teammate or opponent | Edge | One-hot encoding to indicate the relationship between two players. |
| Receiver ID | Global | One-hot encoding to indicate the index of the receiver. |
| Shot indicator | Global | Binary indicator to indicate if there was a threatening shot attempt. |

learning ensures that TacticAI's player representations will be identically computed under such reflections, such that this symmetry does not have to be learnt from data. This proves to be a valuable addition, as high-quality tracking data is often limited—with only a few hundred matches played each year in every league. We provide an in-depth overview of how we employ geometric deep learning in TacticAI in the "Methods" section.

From these representations, TacticAI is then able to answer various predictive questions about the outcomes of a corner—for example, which player is most likely to make first contact with the ball, or whether a shot will take place. TacticAI can also be used as a retrieval system—for mining similar corner kick situations based on the similarity of player representations—and a generative recommendation system, suggesting adjustments to player positions and velocities to maximise or minimise the estimated shot probability. Through several experiments within a case study with domain expert coaches and analysts from Liverpool FC, the results of which we present in the next section, we obtain clear statistical evidence that TacticAI readily provides useful, realistic and accurate tactical suggestions.

## Results

To demonstrate the diverse qualities of our approach, we design TacticAI with three distinct predictive and generative components: receiver prediction, shot prediction, and tactic recommendation through guided generation, which also correspond to the benchmark tasks for quantitatively evaluating TacticAI. In addition to providing accurate quantitative insights for corner kick analysis with its predictive components, the interplay between TacticAI's predictive and generative components allows coaches to sample alternative player setups for each routine of interest, and directly evaluate the possible outcomes of such alternatives.

We will first describe our quantitative analysis, which demonstrates that TacticAI's predictive components are accurate at predicting corner kick receivers and shot situations on held-out test corners and that the proposed player adjustments do not strongly deviate from ground-truth situations. However, such an analysis only gives an indirect insight into how useful TacticAI would be once deployed. We tackle this question of utility head-on and conduct a comprehensive case study in collaboration with our partners at Liverpool FC—where we directly ask human expert raters to judge the utility of TacticAI's predictions and player adjustments. The following sections expand on the specific results and analysis we have performed.

In what follows, we will describe TacticAI's components at a minimal level necessary to understand our evaluation. We defer detailed descriptions of TacticAI's components to the "Methods" section. Note that, all our error bars reported in this research are standard deviations.

## Benchmarking TacticAI

We evaluate the three components of TacticAI on a relevant benchmark dataset of corner kicks. Our dataset consists of 7176 corner kicks from the 2020 to 2021 Premier League seasons, which we randomly shuffle and split into a training (80%) and a test set (20%). As previously mentioned, TacticAI operates on graphs. Accordingly, we represent each corner kick situation as a graph, where each node corresponds to a player. The features associated with each node encode the movements (velocities and positions) and simple profiles (heights and weights) of on-pitch players at the timestamp when the corresponding corner kick was being taken by the attacking kicker (see the "Methods" section), and no information of ball movement was encoded. The graphs are fully connected; that is, for every pair of players, we will include the edge connecting them in the graph. Each of these edges encodes a binary feature, indicating whether the two players are on opposing teams or not. For each task, we generated the relevant dataset of node/edge/graph features and corresponding labels (Tables 1 and 2, see the "Methods" section). The components were then trained separately with their corresponding corner kick graphs. In particular, we only employ a minimal set of features to construct the corner kick graphs, without encoding the movements of the ball nor explicitly encoding the distances between players into the graphs. We used a consistent training-test split for all benchmark tasks, as this made it possible to benchmark not only the individual components but also their interactions.

## Accurate receiver and shot prediction through geometric deep learning

One of TacticAI's key predictive models forecasts the receiver out of the 22 on-pitch players. The receiver is defined as the first player touching the ball after the corner is taken. In our evaluation, all methods used the same set of features (see the "Receiver prediction" entry in Table 1 and the "Methods" section). We leveraged the receiver prediction task to benchmark several different TacticAI base models. Our best-performing model—achieving $0.782 \pm 0.039$ in top-3 test accuracy after 50,000 training steps—was a deep graph attention network[11,12], leveraging geometric deep learning[10] through the use of $D_2$ group convolutions[13]. We supplement this result with a detailed ablation study, verifying that both our choice of base architecture and group convolution yielded significant improvements in the receiver prediction task (Supplementary Table 2, see the subsection "Ablation study" in the "Methods" section). Considering that corner kick receiver prediction is a highly challenging task with many factors that are unseen by our model—including fatigue and fitness levels, and actual ball trajectory—we consider TacticAI's top-3 accuracy to reflect a high level of predictive power, and keep the base TacticAI architecture fixed for subsequent studies. In addition to this

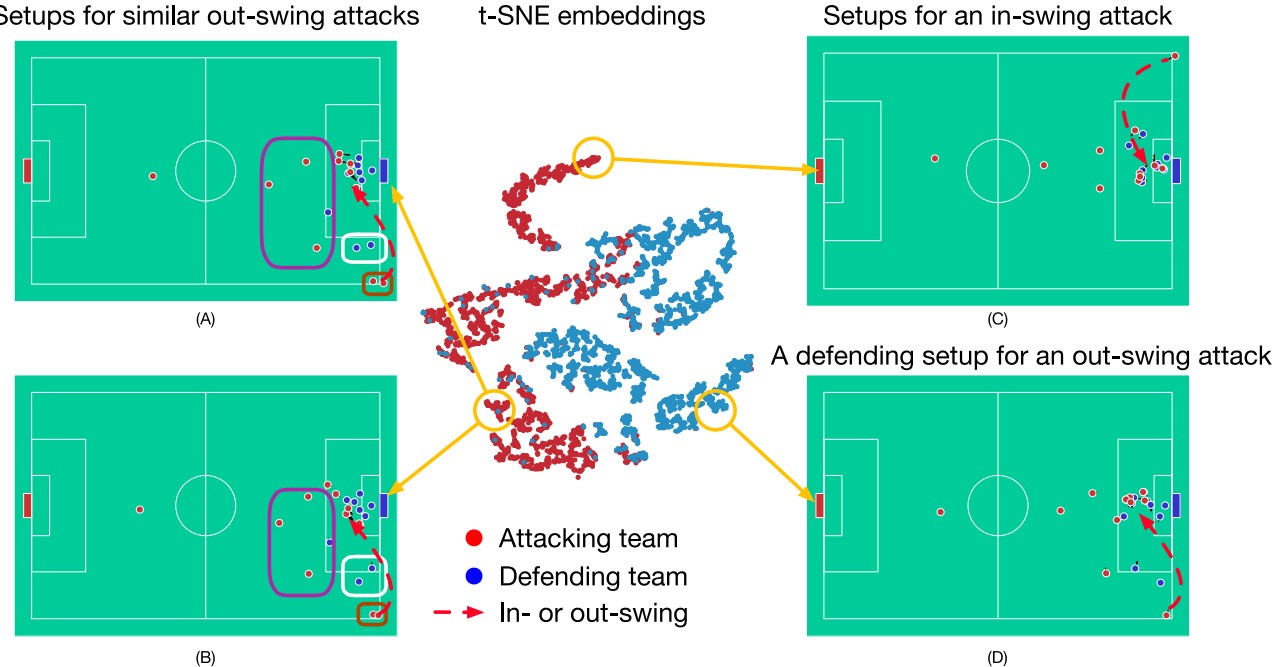

**Fig. 2 | Corner kicks represented in the latent space shaped by TacticAI.** We visualise the latent representations of attacking and defending teams in 1024 corner kicks using *t*-SNE. A latent team embedding in one corner kick sample is the mean of the latent player representations on the same attacking (**A**–**C**) or defending (**D**) team. Given the reference corner kick sample (**A**), we retrieve another corner kick sample (**B**) with respect to the closest distance of their representations in the latent space. We observe that (**A**) and (**B**) are both out-swing corner kicks and share similar patterns of their attacking tactics, which are highlighted with rectangles having the same colours, although they bear differences with respect to the absolute positions and velocities of the players. All the while, the latent representation of an in-swing attack (**C**) is distant from both (**A**) and (**B**) in the latent space. The red arrows are only used to demonstrate the difference between in- and out-swing corner kicks, not the actual ball trajectories.

quantitative evaluation with the evaluation dataset, we also evaluate the performance of TacticAI's receiver prediction component in a case study with human raters. Please see the "Case study with expert raters" section for more details.

For shot prediction, we observe that reusing the base TacticAI architecture to directly predict shot events—i.e., directly modelling the probability $\mathbb{P}(\text{shot}|\text{corner})$—proved challenging, only yielding a test $F_1$ score of $0.52 \pm 0.03$, for a GATv2 base model. Note that here we use the $F_1$ score—the harmonic mean of precision and recall—as it is commonly used in binary classification problems over imbalanced datasets, such as shot prediction. However, given that we already have a potent receiver predictor, we decided to use its output to give us additional insight into whether or not a shot had been taken. Hence, we opted to decompose the probability of taking a shot as

$$\mathbb{P}(\text{shot}|\text{corner}) = \sum_{i \in \text{players}} \mathbb{P}(\text{shot}|\text{receiver}=i, \text{corner})\mathbb{P}(\text{receiver}=i|\text{corner}) \tag{1}$$

where $\mathbb{P}(\text{receiver}|\text{corner})$ are the probabilities computed by TacticAI's receiver prediction system, and $\mathbb{P}(\text{shot}|\text{receiver},\text{corner})$ models the conditional shot probability after a specific player makes first contact with the ball. This was implemented through providing an additional global feature to indicate the receiver in the corresponding corner kick (Table 1) while the architecture otherwise remained the same as that of receiver prediction (Supplementary Fig. 2, see the "Methods" section). At training time, we feed the ground-truth receiver as input to the model—at inference time, we attempt every possible receiver, weighing their contributions using the probabilities given by TacticAI's receiver predictor, as per Eq. (1). This two-phased approach yielded a final test $F_1$ score of $0.68 \pm 0.04$ for shot prediction, which encodes significantly more signal than the unconditional shot

predictor, especially considering the many unobservables associated with predicting shot events. Just as for receiver prediction, this performance can be further improved using geometric deep learning; a conditional GATv2 shot predictor with $D_2$ group convolutions achieves an $F_1$ score of $0.71 \pm 0.01$.

Moreover, we also observe that, even just through predicting the receivers, without explicitly classifying any other salient features of corners, TacticAI learned generalisable representations of the data. Specifically, team setups with similar tactical patterns tend to cluster together in TacticAI's latent space (Fig. 2). However, no clear clusters are observed in the raw input space (Supplementary Fig. 1). This indicates that TacticAI can be leveraged as a useful corner kick retrieval system, and we will present our evaluation of this hypothesis in the "Case study with expert raters" section.

Lastly, it is worth emphasising that the utility of the shot predictor likely does not come from forecasting whether a shot event will occur—a challenging problem with many imponderables—but from analysing the difference in predicted shot probability across multiple corners. Indeed, in the following section, we will show how TacticAI's generative tactic refinements can directly influence the predicted shot probabilities, which will then corresponds to highly favourable evaluation by our expert raters in the "Case study with expert raters" section.

## Controlled tactic refinement using class-conditional generative models

Equipped with components that are able to potently relate corner kicks with their various outcomes (e.g. receivers and shot events), we can explore the use of TacticAI to suggest adjustments of tactics, in order to amplify or reduce the likelihood of certain outcomes.

Specifically, we aim to produce adjustments to the movements of players on one of the two teams, including their positions and

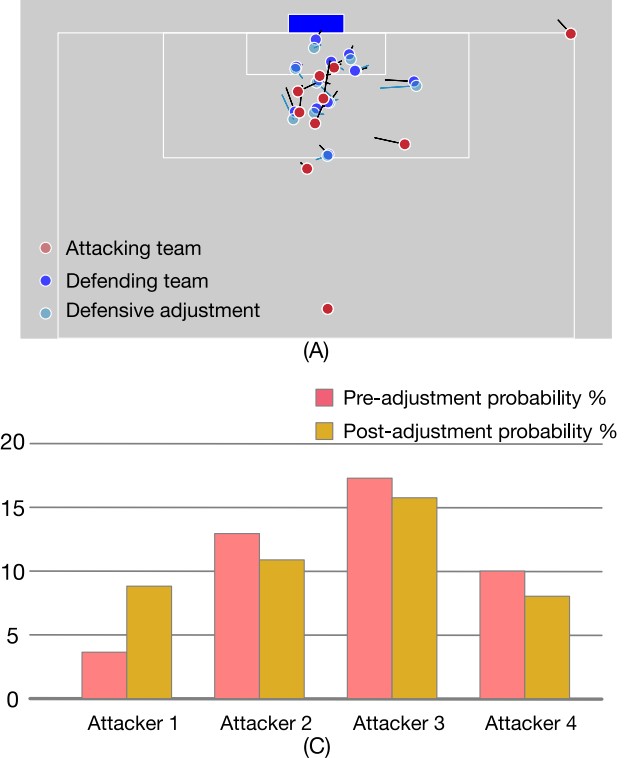

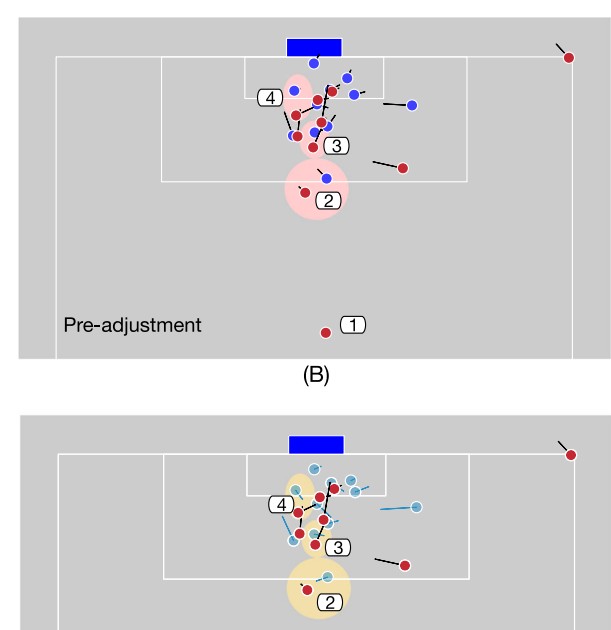

(A) (B) (C) (D)

**Fig. 3 | Example of refining a corner kick tactic with TacticAI.** TacticAI makes it possible for human coaches to redesign corner kick tactics in ways that help maximise the probability of a positive outcome for either the attacking or the defending team by identifying key players, as well as by providing temporally coordinated tactic recommendations that take all players into consideration. As demonstrated in the present example (**A**), for a corner kick in which there was a shot attempt in reality (**B**), TacticAI can generate a tactically-adjusted setting in which the shot probability has been reduced, by adjusting the positioning of the defenders (**D**). The suggested defender positions result in reduced receiver probability for attacking players 2–5 (see bottom row), while the receiver probability of Attacker 1, who is distant from the goalpost, has been increased (**C**). The model is capable of generating multiple such scenarios. Coaches can inspect the different options visually and additionally consult TacticAI's quantitative analysis of the presented tactics.

velocities, which would maximise or minimise the probability of a shot event, conditioned on the initial corner setup, consisting of the movements of players on both teams and their heights and weights. In particular, although in real-world scenarios both teams may react simultaneously to the movements of each other, in our study, we focus on moderate adjustments to player movements, which help to detect players that are not responding to a tactic properly. Due to this reason, we simplify the process of tactic refinement through generating the adjustments for only one team while keeping the other fixed. The way we train a model for this task is through an auto-encoding objective: we feed the ground-truth shot outcome (a binary indicator) as an additional graph-level feature to TacticAI's model (Table 1), and then have it learn to reconstruct a probability distribution of the input player coordinates (Fig. 1B, also see the "Methods" section). As a consequence, our tactic adjustment system does not depend on the previously discussed shot predictor—although we can use the shot predictor to evaluate whether the adjustments make a measurable difference in shot probability.

This autoencoder-based generative model is an individual component that separates from TacticAI's predictive systems. All three systems share the encoder architecture (without sharing parameters), but use different decoders (see the "Methods" section). At inference time, we can instead feed in a desired shot outcome for the given corner setup, and then sample new positions and velocities for players on one team using this probability distribution. This setup, in principle, allows for flexible downstream use, as human coaches can optimise corner kick setups through generating adjustments conditioned on the specific outcomes of their interest—e.g., increasing shot probability for the attacking team, decreasing it for the defending team

(Fig. 3) or amplifying the chance that a particular striker receives the ball.

We first evaluate the generated adjustments quantitatively, by verifying that they are indistinguishable from the original corner kick distribution using a classifier. To do this, we synthesised a dataset consisting of 200 corner kick samples and their corresponding conditionally generated adjustments. Specifically, for corners without a shot event, we generated adjustments for the attacking team by setting the shot event feature to 1, and vice-versa for the defending team when a shot event did happen. We found that the real and generated samples were not distinguishable by an MLP classifier, with an $F_1$ score of $0.53 \pm 0.05$, indicating random chance level accuracy. This result indicates that the adjustments produced by TacticAI are likely similar enough to real corner kicks that the MLP is unable to tell them apart. Note that, in spite of this similarity, TacticAI recommends player-level adjustments that are not negligible—in the following section we will illustrate several salient examples of this. To more realistically validate the practical indistinguishability of TacticAI's adjustments from realistic corners, we also evaluated the realism of the adjustments in a case study with human experts, which we will present in the following section.

In addition, we leveraged our TacticAI shot predictor to estimate whether the proposed adjustments were effective. We did this by analysing 100 corner kick samples in which threatening shots occurred, and then, for each sample, generated one defensive refinement through setting the shot event feature to 0. We observed that the average shot probability significantly decreased, from $0.75 \pm 0.14$ for ground-truth corners to $0.69 \pm 0.16$ for adjustments ($z = 2.62, p < 0.001$). This observation was consistent when testing for attacking

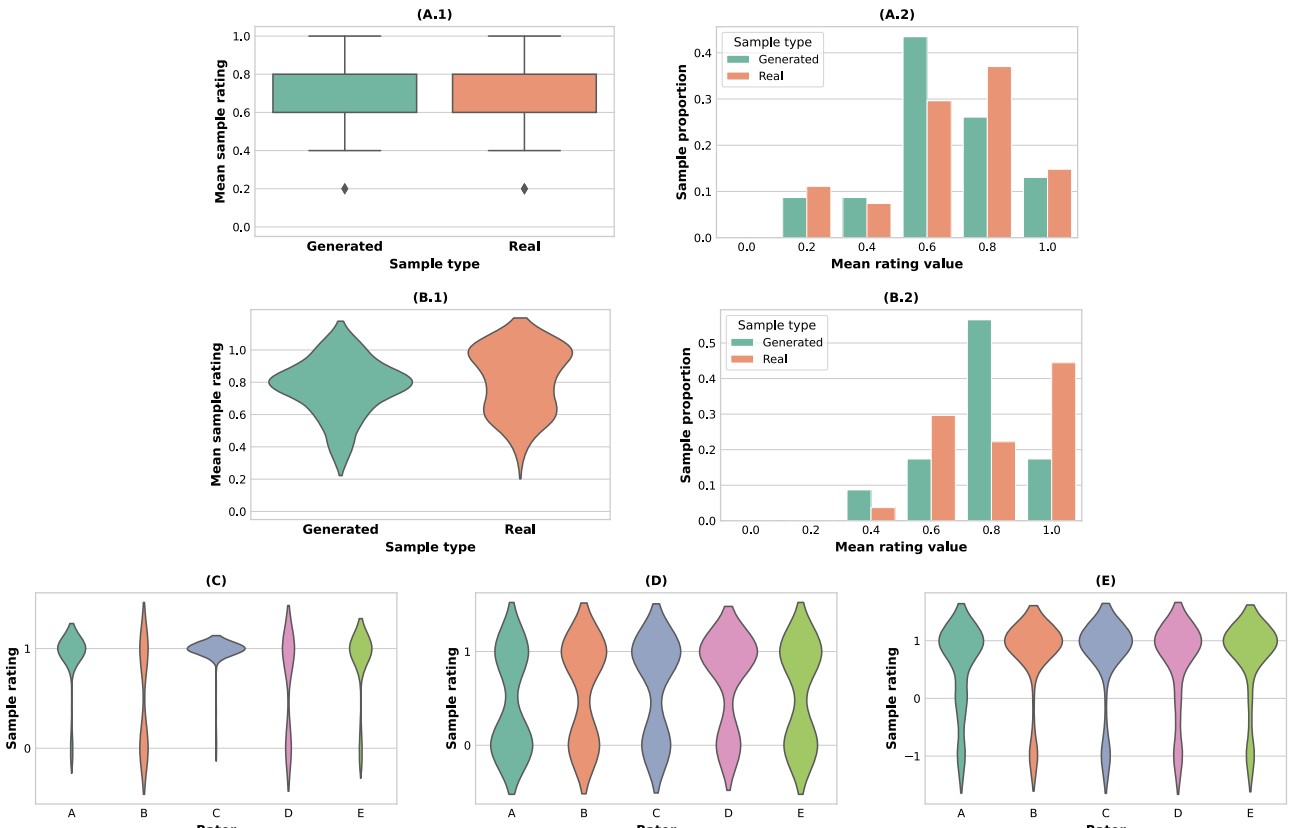

**Fig. 4 | Statistical analysis for the case study tasks.** In task 1, we tested the statistical difference between the real corner kick samples and the synthetic ones generated by TacticAI from two aspects: (**A.1**) the distributions of their assigned ratings, and (**A.2**) the corresponding histograms of the rating values. Analogously, in task 2 (receiver prediction), (**B.1**) we track the distributions of the top-3 accuracy of receiver prediction using those samples, and (**B.2**) the corresponding histogram of the mean rating per sample. No statistical difference in the mean was observed in either cases ((**A.1**) ($z = -0.34, p > 0.05$), and (**B.1**) ($z = 0.97, p > 0.05$)). Additionally, we observed a statistically significant difference between the ratings of different raters on receiver prediction, with three clear clusters emerging (**C**). Specifically, Raters A and E had similar ratings ($z = 0.66, p > 0.05$), and Raters B and D also rated in similar ways ($z = -1.84, p > 0.05$), while Rater C responded differently from all

other raters. This suggests a good level of variety of the human raters with respect to their perceptions of corner kicks. In task 3—identifying similar corners retrieved in terms of salient strategic setups—there were no significant differences among the distributions of the ratings by different raters (**D**), suggesting a high level of agreement on the usefulness of TacticAI's capability of retrieving similar corners ($F_{1,4} = 1.01, p > 0.1$). Finally, in task 4, we compared the ratings of TacticAI's strategic refinements across the human raters (**E**) and found that the raters also agreed on the general effectiveness of the refinements recommended by TacticAI ($F_{1,4} = 0.45, p > 0.05$). Note that the violin plots used in **B.1** and **C**–**E** model a continuous probability distribution and hence assign nonzero probabilities to values outside of the allowed ranges. We only label y-axis ticks for the possible set of ratings.

team refinements (shot probability increased from $0.18 \pm 0.16$ to $0.31 \pm 0.26$ ($z = -4.46, p < 0.001$)). Moving beyond this result, we also asked human raters to assess the utility of TacticAI's proposed adjustments within our case study, which we detail next.

### Case study with expert raters

Although quantitative evaluation with well-defined benchmark datasets was critical for the technical development of TacticAI, the ultimate test of TacticAI as a football tactic assistant is its practical downstream utility being recognised by professionals in the industry. To this end, we evaluated TacticAI through a case study with our partners at Liverpool FC (LFC). Specifically, we invited a group of five football experts: three data scientists, one video analyst, and one coaching assistant. Each of them completed four tasks in the case study, which evaluated the utility of TacticAI's components from several perspectives; these include (1) the realism of TacticAI's generated adjustments, (2) the plausibility of TacticAI's receiver predictions, (3) effectiveness of TacticAI's embeddings for retrieving similar corners, and (4) usefulness of TacticAI's recommended adjustments. We provide an overview of our study's results here and refer the interested reader to Supplementary Figs. 3–5 and the Supplementary Methods for additional details.

We first simultaneously evaluated the realism of the adjusted corner kicks generated by TacticAI, and the plausibility of its receiver predictions. Going through a collection of 50 corner kick samples, we first asked the raters to classify whether a given sample was real or generated by TacticAI, and then they were asked to identify the most likely receivers in the corner kick sample (Supplementary Fig. 3).

On the task of classifying real and generated samples, first, we found that the raters' average $F_1$ score of classifying the real vs. generated samples was only $0.60 \pm 0.04$, with individual $F_1$ scores ($F_1^A = 0.54, F_1^B = 0.64, F_1^C = 0.65, F_1^D = 0.62, F_1^E = 0.56$), indicating that the raters were, in many situations, unable to distinguish TacticAI's adjustments from real corners.

The previous evaluation focused on analysing realism detection performance across raters. We also conduct a study that analyses realism detection across samples. Specifically, we assigned ratings for each sample—assigning +1 to a sample if it was identified as real by a human rater, and 0 otherwise—and computed the average rating for each sample across the five raters. Importantly, by studying the distribution of ratings, we found that there was no significant difference between the average ratings assigned to real and generated corners ($z = -0.34, p > 0.05$) (Fig. 4A). Hence, the real and generated samples

were assigned statistically indistinguishable average ratings by human raters.

For the task of identifying receivers, we rated TacticAI's predictions with respect to a rater as +1 if at least one of the receivers identified by the rater appeared in TacticAI's top-3 predictions, and 0 otherwise. The average top-3 accuracy among the human raters was $0.79 \pm 0.18$; specifically, $0.81 \pm 0.17$ for the real samples, and $0.77 \pm 0.21$ for the generated ones. These scores closely line up with the accuracy of TacticAI in predicting receivers for held-out test corners, validating our quantitative study. Further, after averaging the ratings for receiver prediction sample-wise, we found no statistically significant difference between the average ratings of predicting receivers over the real and generated samples ($z = 0.97$, $p > 0.05$) (Fig. 4B). This indicates that TacticAI was equally performant in predicting the receivers of real corners and TacticAI-generated adjustments, and hence may be leveraged for this purpose even in simulated scenarios.

There is a notably high variance in the average receiver prediction rating of TacticAI. We hypothesise that this is due to the fact that different raters may choose to focus on different salient features when evaluating the likely receivers (or even the amount of likely receivers). We set out to validate this hypothesis by testing the pair-wise similarity of the predictions by the human raters through running a one-away analysis of variance (ANOVA), followed by a Tukey test. We found that the distributions of the five raters' predictions were significantly different ($F_{1,4} = 14.46$, $p < 0.001$) forming three clusters (Fig. 4C). This result indicates that different human raters—as suggested by their various titles at LFC—may often use very different leads when suggesting plausible receivers. The fact that TacticAI manages to retain a high top-3 accuracy in such a setting suggests that it was able to capture the salient patterns of corner kick strategies, which broadly align with human raters' preferences. We will further test this hypothesis in the third task—identifying similar corners.

For the third task, we asked the human raters to judge 50 pairs of corners for their similarity. Each pair consisted of a reference corner and a retrieved corner, where the retrieved corner was chosen either as the nearest-neighbour of the reference in terms of their TacticAI latent space representations, or—as a feature-level heuristic—the cosine similarities of their raw features (Supplementary Fig. 4) in our corner kick dataset. We score the raters' judgement of a pair as +1 if they considered the corners presented in the case to be usefully similar, otherwise, the pair is scored with 0. We first computed, for each rater, the recall with which they have judged a baseline- or TacticAI-retrieved pair as usefully similar—see description of Task 3 in the Supplementary Methods. For TacticAI retrievals, the average recall across all raters was $0.59 \pm 0.09$, and for the baseline system, the recall was $0.36 \pm 0.10$. Secondly, we assess the statistical difference between the results of the two methods by averaging the ratings for each reference–retrieval pair, finding that the average rating of TacticAI retrievals is significantly higher than the average rating of baseline method retrievals ($z = 2.34$, $p < 0.05$). These two results suggest that TacticAI significantly outperforms the feature-space baseline as a method for mining similar corners. This indicates that TacticAI is able to extract salient features from corners that are not trivial to extract from the input data alone, reinforcing it as a potent tool for discovering opposing team tactics from available data. Finally, we observed that this task exhibited a high level of inter-rater agreement for TacticAI-retrieved pairs ($F_{1,4} = 1.01$, $p > 0.1$) (Fig. 4D), suggesting that human raters were largely in agreement with respect to their assessment of TacticAI's performance.

Finally, we evaluated TacticAI's player adjustment recommendations for their practical utility. Specifically, each rater was given 50 tactical refinements together with the corresponding real corner kick setups—see Supplementary Fig. 5, and the "Case study design" section in the Supplementary Methods. The raters were then asked to rate each refinement as saliently improving the tactics (+1), saliently

making them worse (−1), or offering no salient differences (0). We calculated the average rating assigned by each of the raters (giving us a value in the range $[-1, 1]$ for each rater). The average of these values across all five raters was $0.7 \pm 0.1$. Further, for 45 of the 50 situations (90%), the human raters found TacticAI's suggestion to be favourable on average (by majority voting). Both of these results indicate that TacticAI's recommendations are salient and useful to a downstream football club practitioner, and we set out to validate this with statistical tests.

We performed statistical significance testing of the observed positive ratings. First, for each of the 50 situations, we averaged its ratings across all five raters and then ran a $t$-test to assess whether the mean rating was significantly larger than zero. Indeed, the statistical test indicated that the tactical adjustments recommended by TacticAI were constructive overall ($t_{49}^{avg} = 9.20$, $p < 0.001$). Secondly, we verified that each of the five raters individually found TacticAI's recommendations to be constructive, running a $t$-test on each of their ratings individually. For all of the five raters, their average ratings were found to be above zero with statistical significance ($t_{49}^A = 5.84$, $p^A < 0.001$; $t_{49}^B = 7.88$, $p^B < 0.001$; $t_{49}^C = 7.00$, $p^C < 0.001$; $t_{49}^D = 6.04$, $p^D < 0.001$; $t_{49}^E = 7.30$, $p^E < 0.001$). In addition, their ratings also shared a high level of inter-agreement ($F_{1,4} = 0.45$, $p > 0.05$) (Fig. 4E), suggesting a level of practical usefulness that is generally recognised by human experts, even though they represent different backgrounds.

Taking all of these results together, we find TacticAI to possess strong components for prediction, retrieval, and tactical adjustments on corner kicks. To illustrate the kinds of salient recommendations by TacticAI, in Fig. 5 we present four examples with a high degree of inter-rater agreement.

## Discussion

We have demonstrated an AI assistant for football tactics and provided statistical evidence of its efficacy through a comprehensive case study with expert human raters from Liverpool FC. First, TacticAI is able to accurately predict the first receiver after a corner kick is taken as well as the probability of a shot as the direct result of the corner. Second, TacticAI has been shown to produce plausible tactical variations that improve outcomes in a salient way, while being indistinguishable from real scenarios by domain experts. And finally, the system's latent player representations are a powerful means to retrieve similar set-piece tactics, allowing coaches to analyse relevant tactics and counter-tactics that have been successful in the past.

The broader scope of strategy modelling in football has previously been addressed from various individual angles, such as pass prediction[14–16], shot prediction[3] or corner kick tactical classification[7]. However, to the best of our knowledge, our work stands out by combining and evaluating predictive and generative modelling of corner kicks for tactic development. It also stands out in its method of applying geometric deep learning, allowing for efficiently incorporating various symmetries of the football pitch for improved data efficiency. Our method incorporates minimal domain knowledge and does not rely on intricate feature engineering—though its factorised design naturally allows for more intricate feature engineering approaches when such features are available.

Our methodology requires the position and velocity estimates of all players at the time of execution of the corner and subsequent events. Here, we derive these from high-quality tracking and event data, with data availability from tracking providers limited to top leagues. Player tracking based on broadcast video would increase the reach and training data substantially, but would also likely result in noisier model inputs. While the attention mechanism of GATs would allow us to perform introspection of the most salient factors contributing to the model outcome, our method does not explicitly model exogenous (aleatoric) uncertainty, which would be valuable context for the football analyst.

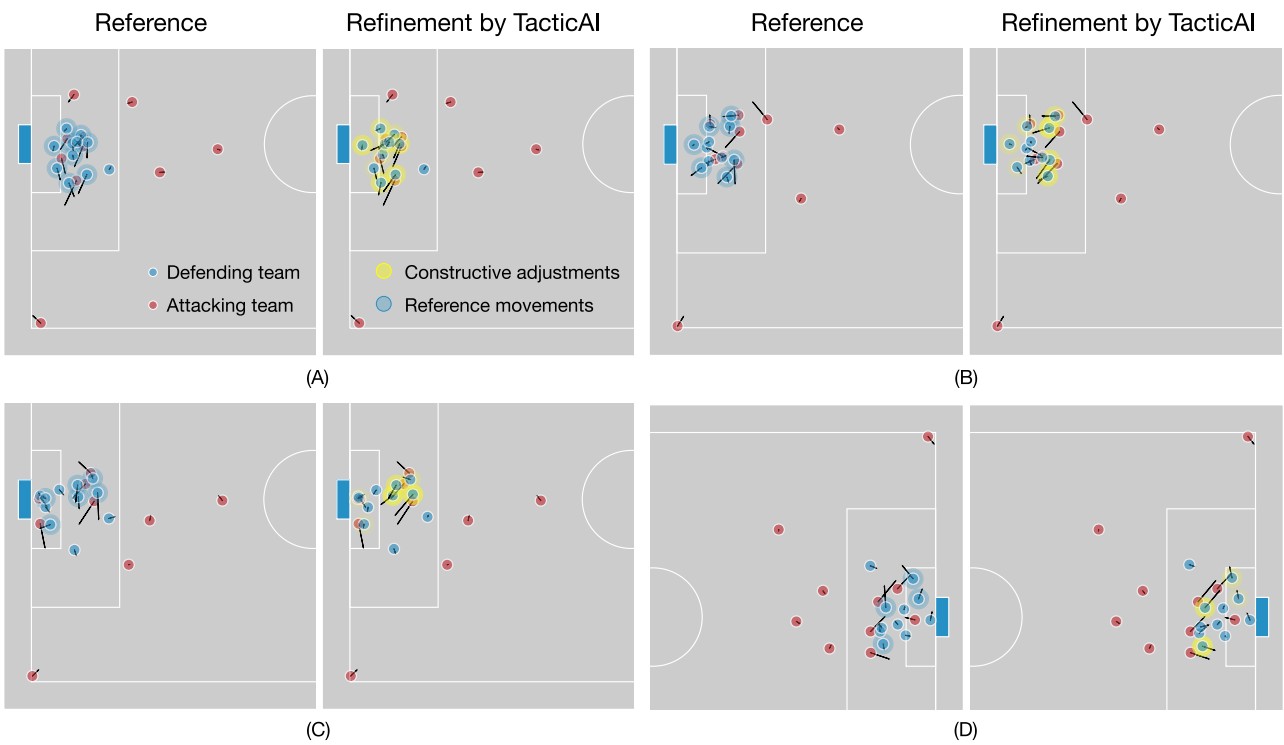

**Fig. 5 | Examples of the tactic refinements recommended by TacticAI.** These examples are selected from our case study with human experts, to illustrate the breadth of tactical adjustments that TacticAI suggests to teams defending a corner. The density of the yellow circles coincides with the number of times that the corresponding change is recognised as constructive by human experts. Instead of optimising the movement of one specific player, TacticAI can recommend improvements for multiple players in one generation step through suggesting better positions to block the opposing players, or better orientations to track them more efficiently. Some specific comments from expert raters follow. In **A**, according to raters, TacticAI suggests more favourable positions for several defenders, and improved tracking runs for several others—further, the goalkeeper is positioned more deeply, which is also beneficial. In **B**, TacticAI suggests that the defenders furthest away from the corner make improved covering runs, which was unanimously deemed useful, with several other defenders also positioned more favourably. In **C**, TacticAI recommends improved covering runs for a central group of defenders in the penalty box, which was unanimously considered salient by our raters. And in **D**, TacticAI suggests substantially better tracking runs for two central defenders, along with a better positioning for two other defenders in the goal area.

While the empirical study of our method's efficacy has been focused on corner kicks in association football, it readily generalises to other set pieces (such as throw-ins, which similarly benefit from similarity retrieval, pass and/or shot prediction) and other team sports with suspended play situations. The learned representations and overall framing of TacticAI also lay the ground for future research to integrate a natural language interface that enables domain-grounded conversations with the assistant, with the aim to retrieve particular situations of interest, make predictions for a given tactical variant, compare and contrast, and guide through an interactive process to derive tactical suggestions. It is thus our belief that TacticAI lays the groundwork for the next-generation AI assistant for football.

## Methods

We devised TacticAI as a geometric deep learning pipeline, further expanded in this section. We process labelled spatio-temporal football data into graph representations, and train and evaluate on benchmarking tasks cast as classification or regression. These steps are presented in sequence, followed by details on the employed computational architecture.

### Raw corner kick data

The raw dataset consisted of 9693 corner kicks collected from the 2020–21, 2021–22, and 2022–23 (up to January 2023) Premier League seasons. The dataset was provided by Liverpool FC and comprises four separate data sources, described below.

Our primary data source is spatio-temporal trajectory frames (tracking data), which tracked all on-pitch players and the ball, for each match, at 25 frames per second. In addition to player positions, their velocities are derived from position data through filtering. For each corner kick, we only used the frame in which the kick is being taken as input information.

Secondly, we also leverage event stream data, which annotated the events or actions (e.g., passes, shots and goals) that have occurred in the corresponding tracking frames.

Thirdly, the line-up data for the corresponding games, which recorded the players' profiles, including their heights, weights and roles, is also used.

Lastly, we have access to miscellaneous game data, which contains the game days, stadium information, and pitch length and width in meters.

### Graph representation and construction

We assumed that we were provided with an input graph $\mathcal{G} = (\mathcal{V}, \mathcal{E})$ with a set of nodes $\mathcal{V}$ and edges $\mathcal{E} \subseteq \mathcal{V} \times \mathcal{V}$. Within the context of football games, we took $\mathcal{V}$ to be the set of 22 players currently on the pitch for both teams, and we set $\mathcal{E} = \mathcal{V} \times \mathcal{V}$; that is, we assumed all pairs of players have the potential to interact. Further analyses, leveraging more specific choices of $\mathcal{E}$, would be an interesting avenue for future work.

Additionally, we assume that the graph is appropriately featurised. Specifically, we provide a node feature matrix, $\mathbf{X} \in \mathbb{R}^{|\mathcal{V}| \times k}$, an edge feature tensor, $\mathbf{E} \in \mathbb{R}^{|\mathcal{V}| \times |\mathcal{V}| \times l}$, and a graph feature vector, $\mathbf{g} \in \mathbb{R}^m$.

The appropriate entries of these objects provide us with the input features for each node, edge, and graph. For example, $\mathbf{x}_u \in \mathbb{R}^k$ would provide attributes of an individual player $u \in \mathcal{V}$, such as position, height and weight, and $\mathbf{e}_{uv} \in \mathbb{R}^l$ would provide the attributes of a particular pair of players $(u, v) \in \mathcal{E}$, such as their distance, and whether they belong to the same team. The graph feature vector, $\mathbf{g}$, can be used to store global attributes of interest to the corner kick, such as the game time, current score, or ball position. For a simplified visualisation of how a graph neural network would process such an input, refer to Fig. 1A.

To construct the input graphs, we first aligned the four data sources with respect to their game IDs and timestamps and filtered out 2517 invalid corner kicks, for which the alignment failed due to missing data, e.g., missing tracking frames or event labels. This filtering yielded 7176 valid corner kicks for training and evaluation. We summarised the exact information that was used to construct the input graphs in Table 2. In particular, other than player heights (measured in centimeters (cm)) and weights (measured in kilograms (kg)), the players were anonymous in the model. For the cases in which the player profiles were missing, we set their heights and weights to 180 cm and 75 kg, respectively, as defaults. In total, we had 385 such occurrences out of a total of 213,246($= 22 \times 9693$) during data preprocessing. We downscaled the heights and weights by a factor of 100. Moreover, for each corner kick, we zero-centred the positions of on-pitch players and normalised them onto a 10 m × 10 m pitch, and their velocities were rescaled accordingly. For the cases in which the pitch dimensions were missing, we used a standard pitch dimension of 110 m × 63 m as default.

We summarised the grouping of the features in Table 1. The actual features used in different benchmark tasks may differ, and we will describe this in more detail in the next section. To focus on modelling the high-level tactics played by the attacking and defending teams, other than a binary indicator for ball possession—which is 1 for the corner kick taker and 0 for all other players—no information of ball movement, neither positions nor velocities, was used to construct the input graphs. Additionally, we do not have access to the player's vertical movement, therefore only information on the two-dimensional movements of each player is provided in the data. We do however acknowledge that such information, when available, would be interesting to consider in a corner kick outcome predictor, considering the prevalence of aerial battles in corners.

## Benchmark tasks construction

TacticAI consists of three predictive and generative models, which also correspond to three benchmark tasks implemented in this study. Specifically, (1) Receiver prediction, (2) Threatening shot prediction, and (3) Guided generation of team positions and velocities (Table 1). The graphs of all the benchmark tasks used the same feature space of nodes and edges, differing only in the global features.

For all three tasks, our models first transform the node features to a latent node feature matrix, $\mathbf{H} = f_{\mathcal{G}}(\mathbf{X}, \mathbf{E}, \mathbf{g})$, from which we could answer queries: either about individual players—in which case we learned a relevant classifier or regressor over the $\mathbf{h}_u$ vectors (the rows of $\mathbf{H}$)—or about the occurrence of a global event (e.g. shot taken)—in which case we classified or regressed over the aggregated player vectors, $\sum_u \mathbf{h}_u$. In both cases, the classifiers were trained using stochastic gradient descent over an appropriately chosen loss function, such as categorical cross-entropy for classifiers, and mean squared error for regressors.

For different tasks, we extracted the corresponding ground-truth labels from either the event stream data or the tracking data. Specifically, (1) We modelled receiver prediction as a node classification task and labelled the first player to touch the ball after the corner was taken as the target node. This player could be either an attacking or defensive player. (2) Shot prediction was modelled as graph classification. In

particular, we considered a next-ball-touch action by the attacking team as a shot if it was a direct corner, a goal, an aerial, hit on the goalposts, a shot attempt saved by the goalkeeper, or missing target. This yielded 1736 corners labelled as a shot being taken, and 5440 corners labelled as a shot not being taken. (3) For guided generation of player position and velocities, no additional label was needed, as this model relied on a self-supervised reconstruction objective.

The entire dataset was split into training and evaluation sets with an 80:20 ratio through random sampling, and the same splits were used for all tasks.

## Graph neural networks

The central model of TacticAI is the graph neural network (GNN)[9], which computes latent representations on a graph by repeatedly combining them within each node's neighbourhood. Here we define a node's neighbourhood, $\mathcal{N}_u$, as the set of all first-order neighbours of node $u$, that is, $\mathcal{N}_u = \{v \mid (v, u) \in \mathcal{E}\}$. A single GNN layer then transforms the node features by passing messages between neighbouring nodes[17], following the notation of related work[10], and the implementation of the CLRS-30 benchmark baselines[18]:

$$\mathbf{h}_u^{(t)} = \phi\left(\mathbf{h}_u^{(t-1)}, \bigoplus_{v \in \mathcal{N}_u} \psi\left(\mathbf{h}_u^{(t-1)}, \mathbf{h}_v^{(t-1)}, \mathbf{e}_{vu}, \mathbf{g}\right)\right) \quad (2)$$

where $\psi : \mathbb{R}^k \times \mathbb{R}^k \times \mathbb{R}^l \times \mathbb{R}^m \to \mathbb{R}^{k'}$ and $\phi : \mathbb{R}^k \times \mathbb{R}^{k'} \to \mathbb{R}^{k'}$ are two learnable functions (e.g. multilayer perceptrons), $\mathbf{h}_u^{(t)}$ are the features of node $u$ after $t$ GNN layers, and $\bigoplus$ is any permutation-invariant aggregator, such as sum, max, or average. By definition, we set $\mathbf{h}_u^{(0)} = \mathbf{x}_u$, and iterate Eq. (2) for $T$ steps, where $T$ is a hyperparameter. Then, we let $\mathbf{H} = f_{\mathcal{G}}(\mathbf{X}, \mathbf{E}, \mathbf{g}) = \mathbf{H}^{(T)}$ be the final node embeddings coming out of the GNN.

It is well known that Eq. (2) is remarkably general; it can be used to express popular models such as Transformers[19] as a special case, and it has been argued that all discrete deep learning models can be expressed in this form[20,21]. This makes GNNs a perfect framework for benchmarking various approaches to modelling player–player interactions in the context of football.

Different choices of $\psi$, $\phi$ and $\bigoplus$ yield different architectures. In our case, we utilise a message function that factorises into an attentional mechanism, $a : \mathbb{R}^k \times \mathbb{R}^k \times \mathbb{R}^l \times \mathbb{R}^m \to \mathbb{R}$:

$$\mathbf{h}_u^{(t)} = \phi\left(\mathbf{h}_u^{(t-1)}, \bigoplus_{v \in \mathcal{N}_u} a\left(\mathbf{h}_u^{(t-1)}, \mathbf{h}_v^{(t-1)}, \mathbf{e}_{vu}, \mathbf{g}\right) \psi\left(\mathbf{h}_v^{(t-1)}\right)\right) \quad (3)$$

yielding the graph attention network (GAT) architecture[12]. In our work, specifically, we use a two-layer multilayer perceptron for the attentional mechanism, as proposed by GATv2[11]:

$$a\left(\mathbf{h}_u^{(t-1)}, \mathbf{h}_v^{(t-1)}, \mathbf{e}_{vu}, \mathbf{g}\right) = \operatorname*{softmax}_{v \in \mathcal{N}_u} \mathbf{a}^\top \operatorname{LeakyReLU}\left(\mathbf{W}_1 \mathbf{h}_u^{(t-1)} + \mathbf{W}_2 \mathbf{h}_v^{(t-1)} + \mathbf{W}_e \mathbf{e}_{vu} + \mathbf{W}_g \mathbf{g}\right)$$

$$(4)$$

where $\mathbf{W}_1, \mathbf{W}_2 \in \mathbb{R}^{k \times h}$, $\mathbf{W}_e \in \mathbb{R}^{l \times h}$, $\mathbf{W}_g \in \mathbb{R}^{m \times h}$ and $\mathbf{a} \in \mathbb{R}^h$ are the learnable parameters of the attentional mechanism, and LeakyReLU is the leaky rectified linear activation function. This mechanism computes coefficients of interaction (a single scalar value) for each pair of connected nodes $(u, v)$, which are then normalised across all neighbours of $u$ using the softmax function.

Through early-stage experimentation, we have ascertained that GATs are capable of matching the performance of more generic choices of $\psi$ (such as the MPNN[17]) while being more scalable. Hence, we focus our study on the GAT model in this work. More details can be found in the subsection "Ablation study" section.

## Geometric deep learning

In spite of the power of Eq. (2), using it in its full generality is often prone to overfitting, given the large number of parameters contained in $\psi$ and $\phi$. This problem is exacerbated in the football analytics domain, where gold-standard data is generally very scarce—for example, in the English Premier League, only a few hundred games are played every season.

In order to tackle this issue, we can exploit the immense regularity of data arising from football games. Strategically equivalent game states are also called transpositions, and symmetries such as arriving at the same chess position through different move sequences have been exploited computationally since the 1960s[22]. Similarly, game rotations and reflections may yield equivalent strategic situations[23]. Using the blueprint of geometric deep learning (GDL)[10], we can design specialised GNN architectures that exploit this regularity.

That is, geometric deep learning is a generic methodology for deriving mathematical constraints on neural networks, such that they will behave predictably when inputs are transformed in certain ways. In several important cases, these constraints can be directly resolved, directly informing neural network architecture design. For a comprehensive example of point clouds under 3D rotational symmetry, see Fuchs et al.[24].

To elucidate several aspects of the GDL framework on a high level, let us assume that there exists a group of input data transformations (symmetries), $\mathfrak{G}$ under which the ground-truth label remains unchanged. Specifically, if we let $y(\mathbf{X}, \mathbf{E}, \mathbf{g})$ be the label given to the graph featurised with $\mathbf{X}, \mathbf{E}, \mathbf{g}$, then for every transformation $\mathfrak{g} \in \mathfrak{G}$, the following property holds:

$$y(\mathfrak{g}(\mathbf{X}), \mathfrak{g}(\mathbf{E}), \mathfrak{g}(\mathbf{g})) = y(\mathbf{X}, \mathbf{E}, \mathbf{g}) \tag{5}$$

This condition is also referred to as $\mathfrak{G}$-invariance. Here, by $\mathfrak{g}(\mathbf{X})$ we denote the result of transforming $\mathbf{X}$ by $\mathfrak{g}$—a concept also known as a group action. More generally, it is a function of the form $\mathfrak{G} \times \mathcal{S} \to \mathcal{S}$ for some state set $\mathcal{S}$. Note that a single group element, $\mathfrak{g} \in \mathfrak{G}$ can easily produce different actions on different $\mathcal{S}$—in this case, $\mathcal{S}$ could be $\mathbb{R}^{|\mathcal{V}| \times k}$ ($\mathbf{X}$), $\mathbb{R}^{|\mathcal{V}| \times |\mathcal{V}| \times l}$ ($\mathbf{E}$) and $\mathbb{R}^m$ ($\mathbf{g}$).

It is worth noting that GNNs may also be derived using a GDL perspective if we set the symmetry group $\mathfrak{G}$ to $S_{|\mathcal{V}|}$, the permutation group of $|\mathcal{V}|$ objects. Owing to the design of Eq. (2), its outputs will not be dependent on the exact permutation of nodes in the input graph.

## Frame averaging

A simple mechanism to enforce $\mathfrak{G}$-invariance, given any predictor $f_{\mathcal{G}}(\mathbf{X}, \mathbf{E}, \mathbf{g})$, performs frame averaging across all $\mathfrak{G}$-transformed inputs:

$$f_{\mathcal{G}}^{\text{inv}}(\mathbf{X}, \mathbf{E}, \mathbf{g}) = \frac{1}{|\mathfrak{G}|} \sum_{\mathfrak{g} \in \mathfrak{G}} f_{\mathcal{G}}(\mathfrak{g}(\mathbf{X}), \mathfrak{g}(\mathbf{E}), \mathfrak{g}(\mathbf{g})) \tag{6}$$

This ensures that all $\mathfrak{G}$-transformed versions of a particular input (also known as that input's orbit) will have exactly the same output, satisfying Eq. (5). A variant of this approach has also been applied in the AlphaGo architecture[25] to encode symmetries of a Go board.

In our specific implementation, we set $\mathfrak{G} = D_2 = \{\text{id}, \leftrightarrow, \updownarrow, \leftrightarrow\updownarrow\}$, the dihedral group. Exploiting $D_2$-invariance allows us to encode quadrant symmetries. Each element of the $D_2$ group encodes the presence of vertical or horizontal reflections of the input football pitch. Under these transformations, the pitch is assumed completely symmetric, and hence many predictions, such as which player receives the corner kick, or takes a shot from it, can be safely assumed unchanged. As an example of how to compute transformed features in Eq. (6), $\leftrightarrow(\mathbf{X})$ horizontally reflects all positional features of players in $\mathbf{X}$ (e.g. the coordinates of the player), and negates the $x$-axis component of their velocity.

## Group convolutions

While the frame averaging approach of Eq. (6) is a powerful way to restrict GNNs to respect input symmetries, it arguably misses an opportunity for the different $\mathfrak{G}$-transformed views to interact while their computations are being performed. For small groups such as $D_2$, a more fine-grained approach can be assumed, operating over a single GNN layer in Eq. (2), which we will write shortly as $\mathbf{H}^{(t)} = g_{\mathcal{G}}(\mathbf{H}^{(t-1)}, \mathbf{E}, \mathbf{g})$. The condition that we need a symmetry-respecting GNN layer to satisfy is as follows, for all transformations $\mathfrak{g} \in \mathfrak{G}$:

$$g_{\mathcal{G}}(\mathfrak{g}(\mathbf{H}^{(t-1)}), \mathfrak{g}(\mathbf{E}), \mathfrak{g}(\mathbf{g})) = \mathfrak{g}(g_{\mathcal{G}}(\mathbf{H}^{(t-1)}, \mathbf{E}, \mathbf{g})) \tag{7}$$

that is, it does not matter if we apply $\mathfrak{g}$ it to the input or the output of the function $g_{\mathcal{G}}$—the final answer is the same. This condition is also referred to as $\mathfrak{G}$-equivariance, and it has recently proved to be a potent paradigm for developing powerful GNNs over biochemical data[24,26].

To satisfy $D_2$-equivariance, we apply the group convolution approach[13]. Therein, views of the input are allowed to directly interact with their $\mathfrak{G}$-transformed variants, in a manner very similar to grid convolutions (which is, indeed, a special case of group convolutions, setting $\mathfrak{G}$ to be the translation group). We use $\mathbf{H}_{\mathfrak{g}}^{(t)}$ to denote the $\mathfrak{g}$-transformed view of the latent node features at layer $t$. Omitting $\mathbf{E}$ and $\mathbf{g}$ inputs for brevity, and using our previously designed layer $g_{\mathcal{G}}$ as a building block, we can perform a group convolution as follows:

$$\mathbf{H}_{\mathfrak{g}}^{(t)} = g_{\mathcal{G}}^{\text{equiv}}(\mathbf{H}_{\mathfrak{g}}^{(t-1)}) = \frac{1}{|\mathfrak{G}|} \sum_{\mathfrak{h} \in \mathfrak{G}} g_{\mathcal{G}}\left(\mathbf{H}_{\mathfrak{h}}^{(t-1)} \parallel \mathbf{H}_{\mathfrak{g}^{-1}\mathfrak{h}}^{(t-1)}\right) \tag{8}$$

Here, $\parallel$ is the concatenation operation, joining the two node feature matrices column-wise; $\mathfrak{g}^{-1}$ is the inverse transformation to $\mathfrak{g}$ (which must exist as $\mathfrak{G}$ is a group); and $\mathfrak{g}^{-1}\mathfrak{h}$ is the composition of the two transformations.

Effectively, Eq. (8) implies our $D_2$-equivariant GNN needs to maintain a node feature matrix $\mathbf{H}_{\mathfrak{g}}^{(t)}$ for every $\mathfrak{G}$-transformation of the current input, and these views are recombined by invoking $g_{\mathcal{G}}$ on all pairs related together by applying a transformation $\mathfrak{h}$. Note that all reflections are self-inverses, hence, in $D_2$, $\mathfrak{g} = \mathfrak{g}^{-1}$.

It is worth noting that both the frame averaging in Eq. (6) and group convolution in Eq. (8) are similar in spirit to data augmentation. However, whereas standard data augmentation would only show one view at a time to the model, a frame averaging/group convolution architecture exhaustively generates all views and feeds them to the model all at once. Further, group convolutions allow these views to explicitly interact in a way that does not break symmetries. Here lies the key difference between the two approaches: frame averaging and group convolutions rigorously enforce the symmetries in $\mathfrak{G}$, whereas data augmentation only provides implicit hints to the model about satisfying them. As a consequence of the exhaustive generation, Eqs. (6) and (8) are only feasible for small groups like $D_2$. For larger groups, approaches like Steerable CNNs[27] may be employed.

## Network architectures

While the three benchmark tasks we are performing have minor differences in the global features available to the model, the neural network models designed for them all have the same encoder–decoder architecture. The encoder has the same structure in all tasks, while the decoder model is tailored to produce appropriately shaped outputs for each benchmark task.

Given an input graph, TacticAI's model first generates all relevant $D_2$-transformed versions of it, by appropriately reflecting the player coordinates and velocities. We refer to the original input graph as the identity view, and the remaining three $D_2$-transformed graphs as reflected views.

Once the views are prepared, we apply four group convolutional layers (Eq. (8)) with a GATv2 base model (Eqs. (3) and (4)) as the $g_{\mathcal{G}}$

function. Specifically, this means that, in Eqs. (3) and (4), every instance of $\mathbf{h}_u^{(t-1)}$ is replaced by the concatenation of $(\mathbf{h}_{\mathfrak{h}}^{(t-1)})_u \parallel (\mathbf{h}_{\mathfrak{g}^{-1}\mathfrak{h}}^{(t-1)})_u$. Each GATv2 layer has eight attention heads and computes four latent features overall per player. Accordingly, once the four group convolutions are performed, we have a representation of $\mathbf{H} \in \mathbb{R}^{4 \times 22 \times 4}$, where the first dimension corresponds to the four views ($\mathbf{H}_{id}, \mathbf{H}_{\leftrightarrow}, \mathbf{H}_{\updownarrow}, \mathbf{H}_{\leftrightarrow\updownarrow} \in \mathbb{R}^{22 \times 4}$), the second dimension corresponds to the players (eleven on each team), and the third corresponds to the 4-dimensional latent vector for each player node in this particular view. How this representation is used by the decoder depends on the specific downstream task, as we detail below.

For receiver prediction, which is a fully invariant function (i.e. reflections do not change the receiver), we perform simple frame averaging across all views, arriving at

$$\mathbf{H}^{node} = \frac{\mathbf{H}_{id} + \mathbf{H}_{\leftrightarrow} + \mathbf{H}_{\updownarrow} + \mathbf{H}_{\leftrightarrow\updownarrow}}{4} \tag{9}$$

and then learn a node-wise classifier over the rows of $\mathbf{H}^{node} \in \mathbb{R}^{22 \times 4}$. We further decode $\mathbf{H}^{node}$ into a logit vector $\mathbf{O} \in \mathbb{R}^{22}$ with a linear layer before computing the corresponding softmax cross entropy loss.

For shot prediction, which is once again fully invariant (i.e. reflections do not change the probability of a shot), we can further average the frame-averaged features across all players to get a global graph representation:

$$\mathbf{h}^{graph} = \frac{1}{22} \sum_{u=1}^{22} \mathbf{h}_u^{node} \tag{10}$$

and then learn a binary classifier over $\mathbf{h}^{graph} \in \mathbb{R}^4$. Specifically, we decode the hidden vector into a single logit with a linear layer and compute the sigmoid binary cross-entropy loss with the corresponding label.

For guided generation (position/velocity adjustments), we generate the player positions and velocities with respect to a particular outcome of interest for the human coaches, predicted over the rows of the hidden feature matrix. For example, the model may adjust the defensive setup to decrease the shot probability by the attacking team. The model output is now equivariant rather than invariant—reflecting the pitch appropriately reflects the predicted positions and velocity vectors. As such, we cannot perform frame averaging, and take only the identity view's features, $\mathbf{H}_{id} \in \mathbb{R}^{22 \times 4}$. From this latent feature matrix, we can then learn a conditional distribution from each row, which models the positions or velocities of the corresponding player. To do this, we extend the backbone encoder with conditional variational autoencoder (CVAE[28,29]). Specifically, for the $u$-th row of $\mathbf{H}_{id}$, $\mathbf{h}_u$, we first map its latent embedding to the parameters of a two-dimensional Gaussian distribution $\mathcal{N}(\mu_u | \sigma_u)$, and then sample the coordinates and velocities from this distribution. At training time, we can efficiently propagate gradients through this sampling operation using the reparameterisation trick[28]: sample a random value $\epsilon_u \sim \mathcal{N}(0,1)$ for each player from the unit Gaussian distribution, and then treat $\mu_u + \sigma_u \epsilon_u$ as the sample for this player. In what follows, we omit edge features for brevity. For each corner kick sample $\mathbf{X}$ with the corresponding outcome $\mathbf{o}$ (e.g. a binary value indicating a shot event), we extend the standard VAE loss[28,29] to our case of outcome-conditional guided generation as

$$\mathcal{L}(\theta, \phi) = -\mathbb{E}_{\mathbf{h}_u \sim q_\theta(\mathbf{h}_u | \mathbf{X}, \mathbf{o})}[\log p_\phi(\mathbf{x}_u | \mathbf{h}_u, \mathbf{o})] + \mathbb{KL}(q_\theta(\mathbf{h}_u | \mathbf{X}, \mathbf{o}) \parallel p(\mathbf{h}_u | \mathbf{o})) \tag{11}$$

where $\mathbf{h}_u$ is the player embedding corresponding to the $u$th row of $\mathbf{H}_{id}$, and $\mathbb{KL}$ is Kullback–Leibler (KL) divergence. Specifically, the first term is the generation loss between the real player input $\mathbf{x}_u$ and the reconstructed sample decoded from $\mathbf{h}_u$ with the decoder $p_\phi$. Using the

KL term, the distribution of the latent embedding $\mathbf{h}_u$ is regularised towards $p(\mathbf{h}_u | \mathbf{o})$, which is a multivariate Gaussian in our case.

A complete high-level summary of the generic encoder–decoder equivariant architecture employed by TacticAI can be summarised in Supplementary Fig. 2. In the following section, we will provide empirical evidence for justifying these architectural decisions. This will be done through targeted ablation studies on our predictive benchmarks (receiver prediction and shot prediction).

## Ablation study

We leveraged the receiver prediction task as a way to evaluate various base model architectures, and directly quantitatively assess the contributions of geometric deep learning in this context. We already see that the raw corner kick data can be better represented through geometric deep learning, yielding separable clusters in the latent space that could correspond to different attacking or defending tactics (Fig. 2). In addition, we hypothesise that these representations can also yield better performance on the task of receiver prediction. Accordingly, we ablate several design choices using deep learning on this task, as illustrated by the following four questions:

Does a factorised graph representation help? To assess this, we compare it against a convolutional neural network (CNN[30]) baseline, which does not leverage a graph representation.

Does a graph structure help? To assess this, we compare against a Deep Sets[31] baseline, which only models each node in isolation without considering adjacency information—equivalently, setting each neighbourhood $\mathcal{N}_u$ to a singleton set $\{u\}$.

Are attentional GNNs a good strategy? To assess this, we compare against a message passing neural network[32], MPNN baseline, which uses the fully potent GNN layer from Eq. (2) instead of the GATv2.

Does accounting for symmetries help? To assess this, we compare our geometric GATv2 baseline against one which does not utilise $D_2$ group convolutions but utilises $D_2$ frame averaging, and one which does not explicitly utilise any aspect of $D_2$ symmetries at all.

Each of these models has been trained for a fixed budget of 50,000 training steps. The test top-$k$ receiver prediction accuracies of the trained models are provided in Supplementary Table 2. As already discussed in the section "Results", there is a clear advantage to using a full graph structure, as well as directly accounting for reflection symmetry. Further, the usage of the MPNN layer leads to slight overfitting compared to the GATv2, illustrating how attentional GNNs strike a good balance of expressivity and data efficiency for this task. Our analysis highlights the quantitative benefits of both graph representation learning and geometric deep learning for football analytics from tracking data. We also provide a brief ablation study for the shot prediction task in Supplementary Table 3.

## Training details

We train each of TacticAI's models in isolation, using NVIDIA Tesla P100 GPUs. To minimise overfitting, each model's learning objective is regularised with an $L^2$ norm penalty with respect to the network parameters. During training, we use the Adam stochastic gradient descent optimiser[33] over the regularised loss.

All models, including baselines, have been given an equal hyperparameter tuning budget, spanning the number of message passing steps ({1, 2, 4}), initial learning rate ({0.0001, 0.00005}), batch size ({128, 256}) and $L^2$ regularisation coefficient ({0.01, 0.005, 0.001, 0.0001, 0}). We summarise the chosen hyperparameters of each TacticAI model in Supplementary Table 1.

## Data availability

The data collected in the human experiments in this study have been deposited in the Zenodo database under accession code https://zenodo.org/records/10557063, and the processed data which is used in the statistical analysis and to generate the relevant

figures in the main text are available under the same accession code. The input and output data generated and/or analysed during the current study are protected and are not available due to data privacy laws and licensing restrictions. However, contact details of the input data providers are available from the corresponding authors on reasonable request.

## Code availability

All the core models described in this research were built with the Graph Neural Network processors provided by the CLRS Algorithmic Reasoning Benchmark[18], and their source code is available at https://github.com/google-deepmind/clrs. We are unable to release our code for this work as it was developed in a proprietary context; however, the corresponding authors are open to answer specific questions concerning re-implementations on request. For general data analysis, we used the following freely available packages: `numpy v1.25.2`, `pandas v1.5.3`, `matplotlib v3.6.1`, `seaborn v0.12.2` and `scipy v1.9.3`. Specifically, the code of the statistical analysis conducted in this study is available at https://zenodo.org/records/10557063.

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

## Acknowledgements

We gratefully acknowledge the support of James French, Timothy Waskett, Hans Leitert and Benjamin Hervey for their extensive efforts in analysing TacticAI's outputs. Further, we are thankful to Kevin McKee, Sherjil Ozair and Beatrice Bevilacqua for useful technical discussions, and Marc Lanctôt and Satinder Singh for reviewing the paper prior to submission.

## Author contributions

Z.W., D. Hennes, L.P. and K.T. coordinated and organised the research effort leading to this paper. P.V. and Z.W. developed the core TacticAI models. Z.W., W.S. and I.G. prepared the Premier League corner kick dataset used for training and evaluating these models. P.V., Z.W., D. Hennes and N.T. designed the case study with human experts and Z.W. and P.V. performed the qualitative evaluation and statistical analysis of its outcomes. Z.W., P.V., D. Hennes, N.T., L.P., M. Kaisers, Y.B., R.E., L.K.W., F.P., W.S., I.G., N.H., M.B., D. Hassabis and K.T. contributed to writing the paper and providing feedback on the final manuscript. J.C., Y.Y., A.R., M. Khan, N.B., P.S. and P.M. contributed valuable technical and implementation discussions throughout the work's development.

## Competing interests

The authors declare no competing interests but the following competing interests: TacticAI was developed during the course of the Authors' employment at Google DeepMind and Liverpool Football Club, as applicable to each Author.
