## [Peer Review File · Nature Communications]

TacticAI: an AI assistant for football tacticsREVIEWER COMMENTS

Reviewer #1 (Remarks to the Author):

The article is concerned with the development of an AI system for football (corner) tactics. The most important features of the AI is that the corner setups are process as (1) graphs in (2) a D2-equivariant manner using group convolutions. The graph structure entails modeling the 22 players as nodes in a complete graph. The nodes are equipped with information about the players positions, velocities, weights and whether they have got the ball or not. The edges are equipped with a feature indicating whether the players are on the same or opposite teams.

The AI is trained on corner kick data to perform three tasks: 1. Prediction of the receiving player 2. Prediction of a shot 3. Suggestion of tactics adjustments. The performance of the AI is extensively benchmarked against human raters. The AI shows a human level performance at predicting the receiver, and is shown to generate both plausible and useful tactic suggestions.

This paper was really interesting to read. Based on the results of the authors study, the approach presented by the paper seems very promising. Although this study focuses on corner data, the way the model is setup seems general enough to also be useable in other situations on the pitch. The choice of restricting to corner kicks as a first example is however a well-motivated one - in short, corner kicks are common and have the least moving parts among significant passages of play(excluding penalty kicks).

The methodology of evaluating the model, as well as its design, is well-described and sound. There are some details that could be explained better - see the list below.

Language-wise, the article is excellent. It is easy to read, and I could only find one type: a "the the" in the first paragraph on page 14. The placements of the tables and figures is sub-optimal, but I expect this to change in the final version.

Now for the concrete questions:

1. How is the ball possession feature defined? More concretely, is it simply one for the corner taker and zero for all others?
2. In the ablation study, which is used to argue the benefit of geometric deep learning, all models are given a fixed budget of 50000 training steps. In order for this to be fair, it is very important that the training rates are set properly (or more blatantly, fine-tuned to the same extent) for all models. This is however not commented on at all. Can the authors do so?
3. It seems like the human raters were not given information about the heights and weights of the players. Why? Is the inclusion of height and weight-data important for the AI model?
4. I have a few questions regarding Figure 4. In Figure (B), the range of possible values (as I understand the text) should be between 0 and 1 - however, the figure shows values between 0.2 and 1.2. In the same vein, the values in figures (C) and (D) should be between 0 and 1, and -1 and 1, respetively, but are instead between -.25 and 1.25 and -.5 and 1.5. Have the numbers been shifted?

Also, the (C)-label appears to have been forgotten.

All in all, I think the paper should be published - the questions above are details which does not invalidate the quality of the paper as a whole.

Reviewer #2 (Remarks to the Author):

The TacticAI system introduced in this paper presents an original AI framework designed to analyze corner kick strategies and tactics in football. It demonstrates noteworthy real-world impacts and offers insights to coaches for developing more effective set pieces. The capabilities of predictive modeling and generative adjustments, highlights the potential of AI, especially geometric deep learning, in advancing strategic coaching. The case study and evaluation with domain experts at Liverpool FC provides strong qualitative evidence that TacticAI's predictions, retrievals and recommendations are realistic and useful to coaches. Additionally, the use of graph-based models to represent games and capture player relationships is a natural and highly effective choice.

As a fan of Steven Gerrard and a GNN enthusiast, I thoroughly enjoyed reading this paper. Below, I have provided two detailed comments and highlighted several minor issues that could benefit from further attention.

1. Firstly, the current configuration of the corner kick graph encompasses a complete graph that includes players from both teams. However, it lacks the inclusion of player team membership as a node feature. To overcome this limitation, it would be worthwhile to explore the utilization of a heterogeneous graph structure capable of effectively capturing interactions among teammates and rivals.

2. Secondly, the graph is subject to robust geometric constraints due to the players' positions being confined within the field and their velocities being limited by human physical capabilities. It is vital to evaluate whether the graph model adequately incorporates these geometric constraints to ensure an accurate representation of the data. While the authors mention the neural network architecture achieving invariant receiver/shot prediction and equivariant guided generation, it would be beneficial to elaborate on how the architecture effectively accounts for the geometric constraints as well.

Furthermore, there are a few minor aspects that require attention:

3. Regarding corner kicks, the vertical movements of players, such as jumps, hold significance in determining the outcomes. It would be beneficial to explore the possibility of incorporating these vertical movements as part of the players' features.

4. In Equation (11), the VAE loss is conditioned on the outcome o . It might be more appropriate to take the expected value with respect to the distribution of $q_{\theta}(h_u \mid X)$ rather than $q_{\theta}(h_u \mid X, o)$.

5. The ablation study provides a comprehensive comparison of the proposed model with CNN and Deep Set, yielding substantial findings. However, it would be intriguing to include a comparison with the Transformer model as well, given its notable performance in spatial-temporal prediction tasks.

6. Additionally, I suggest that the authors consider citing the paper "Who You Play Affects

How You Play: Predicting Sports Performance Using Graph Attention Networks With Temporal Convolution" (<https://arxiv.org/abs/2303.16741>). This paper explores the construction of player interaction graphs and the use of GATv2 for predicting basketball players' performance, which is highly relevant to the present work.

Reviewer #3 (Remarks to the Author):

The present manuscript proposes using geometric deep learning on graphs to provide insights about football games, specifically “corner plays”. The data corresponding to one corner play is represented as a graph of 22 players (nodes) with node, edge and graph features. This featurized graph is processed by a graph attention network (GAT), to make three types of predictions:

1. Predict which player will receive the ball after the corner kick (node classification),
2. Predict whether there will be a shot after the corner kick (graph classification),
3. Propose strategic changes to the players' setup to either increase (attack adaptation) or decrease (defense adaptation) the probability of a shot.

The method can also be used to retrieve similar corner kicks situations. This is because the three predictions share the encoding part of the autoencoder that represents the backbone of the deep learning prediction. The paper is extremely well written, and very easy to follow. This work will definitely be of significance to the field. The methods is sound, and well described. I recommend for publication if the authors can address the points below.

Comparison with the literature

It is difficult to estimate how the techniques used here (geometric deep learning) compare to ones used in the literature. Paragraphs 4 and 5 of the literature review in S1.1 feel too brief: while they mention that other works have attempted to use neural networks and even graph neural networks to determine shoot events or identify patterns in corner kicks, the authors do not describe how the featurized data, neural network architectures, task's target, or even prediction performances differ. It would be interesting to see a discussion of the performances obtained by the proposed manuscript in comparison with performances obtained by other works.

Experiments: Accurate receiver and shot prediction

The authors mention that they use “7, 176 corner kicks from the 2020–2021 Premier League seasons, which we randomly shuffle and split into a training (80%) and a test set (20%).” ([pdf](zotero://open-pdf/library/items/5HYV2ZKB?page=3)) Do the authors perform the 80-20 split according to the soccer games, or the corner kicks themselves? It feels that it is the latter. However, I worried that this might bias the performance of the receiver prediction. Indeed, within a given soccer game, one team may use a consistent strategy (adapted to their opponent). The team may place their players in the same position for several corner kicks, and it might be the same player receiving the ball. Thus, the test set could be somehow contained in the train set and thus bias favorable the prediction accuracy. What do the authors think about this?

The performance of the prediction of the receiver supports the conclusion that “TacticAI’s predictive components are accurate at predicting corner kick receivers”, provided that the split 80-20, on corners and not games, is acceptable.

However, the shot prediction could be more accurate showing a F1 score of 0.64.

Experiments: Controlled tactic refinement

The control tactic refinement relies on the TacticAI being able to predict a dangerous shot. However, we just saw that it is not great at this. This could put into question the quality of the proposed adaptation of the players positions on the field. The study with real soccer experts somewhat alleviates this worry; yet it would be more convincing to see a higher accuracy in the shot prediction task.

On the use of D2 symmetries:

One could argue that the D2 symmetries proposed here are not real symmetries because players may play differently on one side of the field (where their supporters are) than on another, or on the left side (if their strong foot is the left one) compared to the right side. This could be the reason why, in Table 2, adding D2 barely significantly improves the Top-3 accuracy?

Minor comments

“We found that the real and generated samples were not distinguishable by an MLP classifier,” ([pdf](zotero://open-pdf/library/items/5HYV2ZKB?page=5))

How do you know that they are not distinguishable because their configurations are very similar, and not because the MLP is crap?

Several times V should be \mathcal{V}

“for every transformation $g \in G$, the following property holds:” ([pdf](zotero://open-pdf/library/items/5HYV2ZKB?page=12))

The notation $g(X)$ is not defined: the concept of group action should be introduced.

“Once the views are prepared, we apply four group convolutional layers (Equation 8) with a GATv2 base model (Equations 3–4)” ([pdf](zotero://open-pdf/library/items/5HYV2ZKB?page=13))

Is the group convolution performed in the $\psi(hv^{t-1})$?

Is the VAE used a conditional VAE? It’s not cited.

“Given an input graph, TacticAI’s model first generates all relevant D2-transformed versions of it, by appropriately reflecting the player coordinates and velocities.” ([pdf](zotero://open-pdf/library/items/5HYV2ZKB?page=13))

The blueprint of GDL allows practitioners to avoid doing data augmentation. Why is it done

here?

“(A) the distributions of the ratings of the two types of samples, a” ([Wang et al., p. 21](zotero://select/library/items/5DXKGNUW)) ([pdf](zotero://open-pdf/library/items/5HYV2ZKB?page=21))

Fig. 4 (A): the two boxplots look *exactly* the same: is this normal?

Fig. 4 (B): the generated distribution looks bimodal: would the authors have an explanation of why that is the case?

Fig. 4 (C) the “(C)” is missing.

Fig. 5 is somewhat hard to read, esp the top left part.

Reply to Reviewers' comments on TacticAI: an AI assistant for football tactics

Wang, Veličković, Hennes *et al.*

We would like to thank all the Reviewers for their careful and kind assessment of our work—while simultaneously recognising our manuscript as worthy of publication and providing us with a plethora of highly useful constructive comments that has helped us improve the manuscript substantially.

In this document, we survey all of the comments left by the reviewers, along with our responses inlined. We also submit an updated version of our manuscript, with all changes **clearly highlighted in red font**. We are hopeful that these updates and responses properly address all of the Reviewers' comments, and that our work is now suitable for acceptance in Nature Communications. Of course, we remain open and available for any follow-up discussions!

Reviewer #1

Language-wise, the article is excellent. It is easy to read, and I could only find one type: a "the the" in the first paragraph on page 14.

Response: Thank you for recognising the quality of writing in our article! The identified typo has now been fixed.

The placements of the tables and figures is sub-optimal, but I expect this to change in the final version.

Response: We thank the reviewer for bringing up this point. For the purpose of the submission we kept all the figures and tables in separate sections, and will of course gladly work with the Editorial team to optimally place each figure and table in the finished product.

How is the ball possession feature defined? More concretely, is it simply one for the corner taker and zero for all others?

Response: You are absolutely correct, and we have now updated the Methods section to make this explicit.

In the ablation study, which is used to argue the benefit of geometric deep learning, all models are given a fixed budget of 50000 training steps. In order for this to be fair, it is very important that the training rates are set properly (or more blatantly, fine-tuned to the same extent) for all models. This is however not commented on at all. Can the authors do so?

Response: Thank you for bringing this to our attention. We have indeed used identical hyperparameter tuning budgets—spanning learning rate, batch size, number of propagation steps, and weight decay penalty—for all ablated models, and we agree it is important to explicitly state this. We have now updated the Methods section to state hyperparameter ranges tuned for all models.

It seems like the human raters were not given information about the heights and weights of the players. Why? Is the inclusion of height and weight-data important for the AI model?

Response: We thank the reviewer for bringing up this interesting point. Firstly, we did not include this data in the case study inputs primarily because the examples were already reasonably cluttered as-is—requiring our raters to keep track of players’ positions, velocities and the character which uniquely identified them. Adding additional information would have likely made this task significantly more taxing. Secondly, we do believe that TacticAI benefits from the inclusion of such features—as they indicate the player’s physical ability e.g. to execute a header shot—but that the model can still make highly performant predictions without them. To verify this, we performed an appropriate ablation analysis, which is now given in S.Table 4 (receiver prediction) and S.Table 5 (shot prediction). As we expected, excluding height and weight features reduces the predictive power in both cases, but the reduction in mean top-3 accuracy / F_1 score is in the $\sim 2\text{--}3\%$ range. These results are in line with our assumptions: including height/weight features is important, but not critical for TacticAI’s predictive power.

I have a few questions regarding Figure 4. In Figure (B), the range of possible values (as I understand the text) should be between 0 and 1 - however, the figure shows values between 0.2 and 1.2. In the same vein, the values in figures (C) and (D) should be between 0 and 1, and -1 and 1, respectively, but are instead between -.25 and 1.25 and -.5 and 1.5. Have the numbers been shifted? Also, the (C)-label appears to have been forgotten.

Response: We sincerely thank you for bringing up this point to our attention. Your observations are correct—though we highlight that the range in (D) is actually $[0, 1]$, as it measures retrieval rating. The violin plots we used in (B)–(E) model a *continuous probability density*, and hence they will have nonzero probabilities assigned outside of the actual observed ranges.

We have now improved Figure 4’s presentation by not showing y -axis labels for values that are impossible ratings. Further, we have now labelled (C).

Reviewer #2

Firstly, the current configuration of the corner kick graph encompasses a complete graph that includes players from both teams. However, it lacks the inclusion of player team membership as a node feature. To overcome this limitation, it would be worthwhile to explore the utilization of a heterogeneous graph structure capable of effectively capturing interactions among teammates and rivals.

Response: Thank you for bringing up the usage of heterogeneous graphs. We do believe this is a worthwhile baseline to include, and have now added a comparison of our model to a heterogeneous GAT, inspired by [5]. Specifically, in each layer, we separately perform message passing across

intra- and inter-team links, and then aggregate the embeddings from these two layers. The results of our ablation are now provided in S.Table 4 (receiver prediction) and S.Table 5 (shot prediction).

Our results indicate that, for receiver prediction, the heterogeneous GAT is roughly on par with our GATv2 model, while it experiences a significant performance downgrade for shot prediction. Overall, we believe that our result indicates that a more unrestricted message passing may be more beneficial overall for TacticAI, given that we are always processing a fixed, small set of 22 nodes. At such scales, it is typically known that fully-connected graphs tend to be competitive with more hand-crafted heterogeneous approaches.

Secondly, the graph is subject to robust geometric constraints due to the players' positions being confined within the field and their velocities being limited by human physical capabilities. It is vital to evaluate whether the graph model adequately incorporates these geometric constraints to ensure an accurate representation of the data. While the authors mention the neural network architecture achieving invariant receiver/shot prediction and equivariant guided generation, it would be beneficial to elaborate on how the architecture effectively accounts for the geometric constraints as well.

Response: This is a very interesting and important point, and since we do not have explicit guarantees by the TacticAI generator that the predicted positions and velocities will have restricted range, we have now performed relevant analyses to evaluate whether these constraints are broken in practice. Our results are outlined in S1.3 and S. Figure 6.

We find that TacticAI generally recommends subtle changes to both position and velocity, with a slight preference to larger speeds—this is consistent with TacticAI being able to detect players that are not following the required strategies, as identified by our expert raters. Concretely, across a diverse set of TacticAI adjustments, we find that *no players* are predicted outside the pitch, if they hadn't already been outside the pitch in the real corner input. Further, we find that only 0.3% of players have recommended speeds that exceed the maximal observed speeds in the dataset.

We generally conclude that TacticAI is able to preserve plausible ranges for its generative outputs. Coupled with the positive realism ratings assigned to TacticAI adjustments by human raters, and the fact exact speeds are not as relevant to coaching decisions, we find strong evidence that TacticAI appropriately satisfies geometric constraints in practice.

Regarding corner kicks, the vertical movements of players, such as jumps, hold significance in determining the outcomes. It would be beneficial to explore the possibility of incorporating these vertical movements as part of the players' features.

Response: This is a very interesting point, which we agree with in principle. However, the tracking dataset we are leveraging does not expose z -axis movements, making us unable to leverage such information in TacticAI at present. We have now updated the Methods section to explicitly call this prospect out.

In Equation (11), the VAE loss is conditioned on the outcome o . It might be more appropriate to take the expected value with respect to the distribution of $q_{\theta}(h_u | X)$ rather than $q_{\theta}(h_u | X, o)$.

Response: We thank the reviewer for the detailed observation. The VAE loss is actually an adaptation from the standard Conditional VAE [3]. Specifically, for the graph input of each corner kick sample, the conditional outcome o is represented as a global feature (See Table 1), whilst X only

represents the player node features. The latent distribution q_θ is also conditioned on the outcome o , and therefore, it may be more appropriate to calculate the expectation of the reconstruction loss with respect to the conditional latent distribution $q_\theta(h_u | X, o)$.

The ablation study provides a comprehensive comparison of the proposed model with CNN and Deep Set, yielding substantial findings. However, it would be intriguing to include a comparison with the Transformer model as well, given its notable performance in spatial-temporal prediction tasks.

Response: We sincerely thank the reviewer for bringing up this point. Much like your point about heterogeneous graphs, we do believe this is a worthwhile baseline to include, and have now added a comparison of our model to a variant using Transformer self-attention [4, 2]. The results of our ablation are now provided in S.Table 4 (receiver prediction) and S.Table 5 (shot prediction).

Our results indicate no significant differences between Transformer and GATv2 attention. We find result unsurprising, given that both of these attentional mechanisms are known to be expressive [1], and the small number of nodes for the player graphs considered.

Additionally, I suggest that the authors consider citing the paper "Who You Play Affects How You Play: Predicting Sports Performance Using Graph Attention Networks With Temporal Convolution" (<https://arxiv.org/abs/2303.16741>). This paper explores the construction of player interaction graphs and the use of GATv2 for predicting basketball players' performance, which is highly relevant to the present work.

Response: We thank the reviewer for this inclusion suggestion. Our updated related work section now includes a citation to this relevant piece of work. Our updated related work section has been further expanded by extending the comparative discussion between our proposed approach and the most similar use case covered in literature, to better highlight the similarities and the differences to the readers. In terms of additional citations, we have also included the following papers in the updated section: "Graph representations for the analysis of multi-agent spatiotemporal sports data" (<https://link.springer.com/article/10.1007/s10489-022-03631-z>) and "Analysis of contextualized intensity in Men's elite handball using graph-based deep learning" (<https://www.tandfonline.com/doi/full/10.1080/02640414.2023.2268366>).

Reviewer #3

It is difficult to estimate how the techniques used here (geometric deep learning) compare to ones used in the literature. Paragraphs 4 and 5 of the literature review in S1.1 feel too brief: while they mention that other works have attempted to use neural networks and even graph neural networks to determine shoot events or identify patterns in corner kicks, the authors do not describe how the featurized data, neural network architectures, task's target, or even prediction performances differ. It would be interesting to see a discussion of the performances obtained by the proposed manuscript in comparison with performances obtained by other works.

Response: Thank you for your feedback regarding our literature review, and we fully agree that it is useful to make our exposition of related work clearer. We would like to start by making it clear

that, *to the best of our knowledge, no prior works have used graph neural networks to model corner kicks*, and we make this explicit now in the literature review.

To make it easier for future readers to understand how the approach proposed in the paper relates to prior work that we cover in literature review in S1.1, we have now significantly expanded the discussion in that section, as per your suggestion, to better contrast the approaches that were originally mentioned in Paragraphs 4 and 5 with the approach taken in the paper. In the discussion we focus on methodological comparisons and differences in features/inputs and use cases. The performance numbers are not directly comparable due to differences in use case definitions (e.g. predicting the probability of a shot during general play within 10 seconds, conditioned on the ball being passed to an attacker vs the probability of a shot coming from a corner kick). On top of the extended discussion, we have also extended S1.1 with multiple additional citations to help the readers contextualize prior approaches for similar use cases in sports.

The authors mention that they use “7, 176 corner kicks from the 2020–2021 Premier League seasons, which we randomly shuffle and split into a training (80%) and a test set (20%).” Do the authors perform the 80-20 split according to the soccer games, or the corner kicks themselves? It feels that it is the latter. However, I worried that this might bias the performance of the receiver prediction. Indeed, within a given soccer game, one team may use a consistent strategy (adapted to their opponent). The team may place their players in the same position for several corner kicks, and it might be the same player receiving the ball. Thus, the test set could be somehow contained in the train set and thus bias favorable the prediction accuracy. What do the authors think about this?

Response: We thank the reviewer for raising a very important point! The split we used in the paper was indeed a fully randomised split per-corner. We found this to be a reasonable assumption as, even though the same strategy may be executed within a single game, its actual unfolding may strongly depend on many factors which are variable throughout a game (such as fatigue). Further, in practice, it is never the case that players will start the corner in *exactly* the same positions.

That being said, we concur that a per-corner split could bias the results compared to a per-game split, and we find it important to investigate this effect. For a fully realistic scenario, we retrained and tested our models using a 80–20 *temporal split*—the test corners are from *the most recent 20% of the games*. These results are now provided in amended S.Table 2 (for receiver prediction) and S.Table 3 (for shot prediction), and discussed in S1.3.

The key findings from our results are:

- *There exists a minor bias from the per-corner split in the receiver prediction task.* All of our methods have reduced mean, and increased variance, when evaluated over a temporal split. However, this reduction is *minor*—under 3% for our best-performing base model, still keeping our model’s accuracy close to the measured human-level performance.
- *There is no strong evidence of bias from the per-corner split in the shot prediction task.* For our best-performing conditional shot predictor, there is no observed reduction in the mean F_1 score when evaluating under a temporal split. Weaker models—e.g. conditional GATv2 without D_2 symmetry—do exhibit an increase in instability, however.
- *The relative performance order of baselines is unchanged.* That is, even over a temporal split, we find a clear advantage to using graph machine learning, as well as geometric deep learning.

- *Geometric models are more stable under the temporal split.* That is, the reduction of the mean accuracy experienced by our D_2 -group convolutional receiver predictor ($< 3\%$) is smaller than the one experienced by non-geometric GNNs ($\geq 5\%$). Further, on shot prediction, we actually observe a slight *increase* in the mean for the geometric model. We suspect that the increased stability is due to the fact that our model’s symmetry enforces that its results will remain consistent for identical but symmetrical situations, and thus it will be overall less sensitive to repetitions of similar strategies.

All of these points taken into account, we find the addition of the temporal split to be important, but it does not change our conclusions about TacticAI’s strength, or the utility of its various components like D_2 symmetry.

However, the shot prediction could be more accurate showing a F1 score of 0.64.

Response: We thank the reviewer for inviting our discussion on the performance of our shot predictor. We would like to start by remarking that we mistranscribed this number from our ablation table in Supplementary materials—the conditional F_1 score of our GATv2 base model is 0.68 ± 0.04 , and it can be further improved to 0.71 ± 0.01 by using D_2 convolutions. This has now been fixed in the main paper, and we apologise for the oversight.

Our intuition is that it is likely *quite hard* to expect a very high F_1 score on shot prediction – the final shot event depends on way too many unknowns that our model simply does not have access to. In our opinion, the utility of the shot predictor comes *not* from predicting exactly whether a shot event will occur, but *from analysing relative shot probabilities across several situations*. Indeed, our generative adjustments saliently change the predicted shot probabilities, which in turn correspond to highly favourable evaluation by our expert raters. We have now updated the main paper to reflect this discussion.

The control tactic refinement relies on the TacticAI being able to predict a dangerous shot. However, we just saw that it is not great at this. This could put into question the quality of the proposed adaptation of the players positions on the field. The study with real soccer experts somewhat alleviates this worry; yet it would be more convincing to see a higher accuracy in the shot prediction task.

Response: Our response above already indicates why we do not require higher accuracy in shot prediction to have salient tactical adjustments. Further, we also need to make a slight correction in the reviewer’s summary here. While we use the shot prediction model to quantitatively *evaluate* the saliency of the generated refinements, we *do not use it* within the refinement system itself, so there is no explicit reliance on the shot predictor. Rather, the refinement system is conditioned on a *shot event flag*, which is an *input feature* provided by us, not a label. We have now updated the main paper to make this point clearer.

One could argue that the D_2 symmetries proposed here are not real symmetries because players may play differently on one side of the field (where their supporters are) than on another, or on the left side (if their strong foot is the left one) compared to the right side. This could be the reason why, in Table 2, adding D_2 barely significantly improves the Top-3 accuracy?

Response: Thank you for remarking the precision of the symmetries. We agree that this is a valuable discussion to have. We will first directly address the point about the accuracy improvements of incorporating D_2 symmetry. Then, we will provide some general remarks on footedness in corner kicks, and quantify to what extent D_2 can be considered a symmetry of corner tactics—by providing concrete data from the Premier League.

Firstly, we assert that *the improvements of adding D_2 symmetries are significant*, and we consider this to be the most important piece of evidence that including those symmetries in the model is beneficial. Specifically, when comparing our best D_2 -symmetric model (GATv2 with group convolutions) and our best non- D_2 -symmetric model (base GATv2), adding symmetries improves mean top-3 accuracy by 3.4% (78.2% vs. 74.8%). This jump is not only significant, it is comparable to the kind of improvement experienced by incorporating the graph structure (base GATv2 is 3.5% more accurate than Deep Sets in terms of mean top-3 accuracy). Further, these benefits are persistent in the shot prediction task, and remain more stable under the aforementioned temporal split.

Generally, about 30% of the players in the Premier League are left-footed or both-footed, which gives a sufficient amount of players who can effectively attack a ball in a mirrored capacity. As an example, most teams have a left-footed left-back and a right-footed right-back who take corners, and it would be rare to find a team that doesn't have both a left-footed and right-footed corner taker. In terms of attacking the corners, headed shots are very common in corner kick situations, so footedness tends to matter significantly less in these situations. Also, many shots taken during corners are more reactive from close range, where foot preference matters less.

To offer a concrete, measurable data point to substantiate this, we studied how symmetrical *shot events* are with respect to D_2 symmetries. Our data spans all corner kicks taken in the Premier League spanning the 2020/21, 2021/22, and 2022/23 seasons. Our analysis shows that the percentage of shots taken from left-footed corners taken from the left side (20%) are roughly equal to shots taken from right-footed corners from the right side (22%). Similarly, we observe a similar percentage of shots taken by left-footed corners from the right side (15%) and right-footed corners from the left side (14%). We believe this data point improves confidence that D_2 can be considered a useful approximate symmetry of corner kick outcomes.

“We found that the real and generated samples were not distinguishable by an MLP classifier,” How do you know that they are not distinguishable because their configurations are very similar, and not because the MLP is crap?

Response: We thank the reviewer for raising this point. We cannot say this with certainty, but we do believe that the most likely reason for the MLP's underperformance is the fact the configurations are very similar. Note that, in spite of this similarity, TacticAI recommends player-level adjustments that are *not negligible*, with Figure 5 providing several salient examples. We have now revised the main paper to explicitly call both of these points out. Further, we would like to point out that, in either case, we assign a far greater value to the human expert evaluation of realism—which appears to validate the quantitative observations in the MLP experiment.

Several times V should be \mathcal{V}

Response: Thank you for noticing this—we now corrected all the appearances of V to \mathcal{V} .

“for every transformation $\mathfrak{g} \in \mathfrak{G}$, the following property holds:” The notation $\mathfrak{g}(\mathbf{X})$ is not defined: the concept of group action should be introduced.

Response: Thank you for remarking about this. We agree that it will be useful to more explicitly elaborate on group actions, and we have now done so.

“Once the views are prepared, we apply four group convolutional layers (Equation 8) with a GATv2 base model (Equations 3-4)” Is the group convolution performed in the $\psi(\mathbf{h}_v^{(t-1)})$?

Response: The group convolution is performed in all of the parts of the GATv2 equation (both ψ and a)—all of the $\mathbf{h}_v^{(t-1)}$ inputs are now concatenated across the two relevant views in each invocation of g_G . We have now updated the text to make this explicit.

Is the VAE used a conditional VAE? It’s not cited.

Response: Thank you for noticing this. We indeed adapted the standard conditional VAE (CVAE) to our case of outcome-conditional guided generation. We have now cited the original CVAE paper.

“Given an input graph, TacticAI’s model first generates all relevant D_2 -transformed versions of it, by appropriately reflecting the player coordinates and velocities.” The blueprint of GDL allows practitioners to avoid doing data augmentation. Why is it done here?

Response: We thank the reviewer for raising this point, and it is worthwhile to delimit our approach from traditional data augmentation. Namely, while standard data augmentation would only show one view at a time to the model, a frame averaging/group convolution architecture *exhaustively generates all views* and feeds them to the model *all at once*. The reason why we go for the exhaustive approach here is because it is conceptually simple, and still feasible for *small groups* like D_2 . We have updated the paper in the Methods section to explicitly discuss this, and leave a pointer to approaches like Steerable CNNs, which may be employed when the group size is prohibitively large.

“(A) the distributions of the ratings of the two types of samples, a”

*Fig. 4 (A): the two boxplots look *exactly* the same: is this normal?*

Fig. 4 (B): the generated distribution looks bimodal: would the authors have an explanation of why that is the case?

Fig. 4 (C) the “(C)” is missing.

Response: We sincerely thank the reviewer for bringing up these points to our attention. We have updated Figure 4, including:

- We have updated the ticks of the y -axis of each sub-figure to align with the concrete possible values of the ratings in each task.
- In addition to the box plots in Figure A.1 (previous Fig.A), we have added the corresponding histogram with respect to the values of the ratings in Figure (A.2). We believe that this clarifies the differences between the two distributions, and allows for more easily comparing them.

- For the original Figure B (now Figure B.1), we have fixed an error of the order of the sample types in the x -axis. The green plot now corresponds to the distribution of the generated samples, and the orange one corresponds to that of the real samples. Although this does not change the results of the statistical test, we apologize for any potential confusions.
- In addition to improving the original violin plots, to address your question that the distribution of the real samples (orange) looks bimodal, we have provided the corresponding histogram of the actual values of the ratings in Figure B.2. As can be seen, the ratings of real samples (orange in both Figures B.1 and B.2) have the highest probability mass around the value 1.0. As for the other observed mode (0.6), we think that this is a bias caused by sample size, because the sample proportions of rating=0.6 and rating=0.8 are actually very close (0.28 and 0.24 respectively) but are much smaller than the sample size of rating=1.0 (0.44). Therefore, it is still reasonable to approximate this as a unimodal distribution.
- The missing label for (C) is now reintroduced.

Fig. 5 is somewhat hard to read, esp the top left part.

Response: We thank the reviewer for raising this point. We have improved the visualization of Figure 5 and made the text larger.

References

- [1] Shaked Brody, Uri Alon, and Eran Yahav. How attentive are graph attention networks? *arXiv preprint arXiv:2105.14491*, 2021.
- [2] Luis Müller, Mikhail Galkin, Christopher Morris, and Ladislav Rampásek. Attending to graph transformers, 2023.
- [3] Kihyuk Sohn, Honglak Lee, and Xinchen Yan. Learning structured output representation using deep conditional generative models. In C. Cortes, N. Lawrence, D. Lee, M. Sugiyama, and R. Garnett, editors, *Advances in Neural Information Processing Systems*, volume 28. Curran Associates, Inc., 2015.
- [4] Ashish Vaswani, Noam Shazeer, Niki Parmar, Jakob Uszkoreit, Llion Jones, Aidan N Gomez, Lukasz Kaiser, and Illia Polosukhin. Attention is all you need. *Advances in neural information processing systems*, 30, 2017.
- [5] Xiao Wang, Houye Ji, Chuan Shi, Bai Wang, Yanfang Ye, Peng Cui, and Philip S Yu. Heterogeneous graph attention network. WWW '19, page 2022–2032, New York, NY, USA, 2019. Association for Computing Machinery.

REVIEWERS' COMMENTS

Reviewer #1 (Remarks to the Author):

The authors have taken great care to appropriately respond to both mine and, in my opinion, the other referees' questions and suggestions. I recommend publication.

As a football enthusiast, I cannot resist the urge to remark that I would have responded differently to the remark about vertical information. I watch football quite a lot, and my amateur take would be that it is very uncommon for players to jump *as the corner is taken*, so that it for this special task would not be very interesting. The ability to process the information is of course in general interesting, e.g. for analyzing free kicks (with a wall that can decide to jump or not to jump, etc.), so the formulation in the manuscript is not wrong!

Reviewer #2 (Remarks to the Author):

I would like to thank the authors for the diligent efforts they have made to address the concerns and suggestions raised before. The inclusion of a more comprehensive comparison analysis with the heterogeneous graph attention network is useful. Furthermore, the authors have conducted an ablation study utilizing transformers. The analysis of the prediction results under geometric constraint is also important to showcase the value of the model.

Overall, the revisions have substantively improved the manuscript, and providing convincing evidence that supports the authors' claims. I recommend the paper for publication and look forward to seeing its impact on further research in the area of graph neural networks and sports analysis.

Reviewer #3 (Remarks to the Author):

I thank the authors for carefully addressing my comments. I recommend the paper for publication.

Response to Referees Letter

We sincerely appreciate the constructive and insightful feedback from all the referees, which has helped us improve our manuscript substantially.